# Adsorption-Membrane Hybrid Approach for the Removal of Azithromycin from Water: An Attempt to Minimize Drug Resistance Problem

**Muhammad Wahab** [1], **Muhammad Zahoor** [2,*], **Syed Muhammad Salman** [1], **Abdul Waheed Kamran** [3], **Sumaira Naz** [2], **Juris Burlakovs** [4], **Anna Kallistova** [5], **Nikolai Pimenov** [5] **and Ivar Zekker** [6,*]

1   Department of Chemistry, Islamia College University, Peshawar 25120, Pakistan; mwahabbajaur@gmail.com (M.W.); salman@icp.edu.pk (S.M.S.)
2   Department of Biochemistry, University of Malakand, Chakdara 18800, Pakistan; sumaira.biochem@gmail.com
3   Department of Chemistry, University of Malakand, Chakdara 18800, Pakistan; waheedkamran1989@gmail.com
4   Institute of Forestry and Rural Engineering, Estonian University of Life Sciences, 5 Kreutzwaldi St., 51014 Tartu, Estonia; Juris.burlakovs@emu.ee
5   Winogradsky Institute of Microbiology, Research Centre of Biotechnology of the Russian Academy of Sciences, Leninsky Prospect, 33, Build. 2, 119071 Moscow, Russia; kallistoanna@mail.ru (A.K.); npimenov@mail.ru (N.P.)
6   Institute of Chemistry, Faculty of Science, University of Tartu, 14 Ravila st, 50411 Tartu, Estonia
\*   Correspondence: mohammadzahoorus@yahoo.com (M.Z.); ivar.zekker@ut.ee (I.Z.)

**Abstract:** In this study, activated carbon (AC) and magnetic activated carbon (MAC) were prepared from *Dalbergia sissoo* sawdust for the removal of antibiotic Azithromycin (AZM) from aqueous solution. The effect of initial concentration, contact time, pH, adsorbent dosage, and the temperature were investigated for both the adsorbents. The optimum AZM concentration, contact time, pH and adsorbents dosages were found to be 80 mg/L, 120 min, 6 and 7 (pH, respectively, for AC and MAC), and 0.1 g (for both AC and MAC), respectively. The isothermal data of both sets of experiments correlated well with the Langmuir isotherm model, while the kinetic data with the pseudo-second-order model. The adsorption of AZM on both adsorbents was found to be favorable, which is evident in the values of the thermodynamic parameters ($\Delta H$ = −26.506 and −24.149 KJ/mol, $\Delta S$ = 91.812 and 81.991 J/mol K, respectively, for AC and MAC). To evaluate the effect of AC and MAC on the membrane parameters, a continuous stirred reactor was connected with ultrafiltration (UF), nanofiltration (NF), and reverse osmosis (RO) membranes. High % retention and improved permeate flux (around 90%) were obtained for AC/UF, AC/NF AC/RO, MAC/UF, MAC/NF, and MAC/RO treatments. The percent retention of AZM observed for AC/UF, AC/NF AC/RO was higher than MAC/UF, MAC/NF, and for MAC/RO hybrid processes due to greater surface area of AC than MAC.

**Keywords:** activated carbon; magnetic activated carbon; azithromycin; percent retention; permeate flux

## 1. Introduction

Pharmaceutical products have been frequently found in the environment and water bodies, treated wastewater, and in potable water [1]. Azithromycin is a broad-spectrum antibiotic with bacteriostatic activity against many Gram-positive and Gram-negative bacteria. The half-life of the antibiotic is 2–4 days and it is predominantly eliminated from the body through fecal and urinary excretion [2]. The excreted antibiotics enter into the environment through wastewater [3]. The occurrence of these antibiotics in the environment may cause negative impacts on aquatic and terrestrial ecosystems [4,5].

Therefore, the removal of such antibiotics from wastewater to avoid release into the aquatic environment is essential to stop the propagation of antibiotic-resistant bacteria species.

Various methods have been developed for the removal of antibiotics such as precipitation, ion exchange, flotation, electrochemical techniques, adsorption, and membrane hybrid technology [6]. Among these methods, the adsorption of antibiotics has been proven to be an effective one [7]. Different adsorptive substances, such as natural clay materials such as zeolite, clay, silica gel, and bentonite, in addition to activated carbon and carbon nanotubes, such as multi-walled carbon nanotubes, have been applied for the removal of antibiotics from wastewater [8–10].

Nevertheless, while dealing with a large amount of water, treatment through these adsorbents may make the process expensive [11]. Correspondingly, treatment procedures involving membrane separation processes such as microfiltration (MF), ultrafiltration (UF), nanofiltration (NF), or reverse osmosis (RO) are also employed, because they are environment-friendly and have a high separation efficiency [12–14]. Research on the production of membranes in recent years has resulted in the reduction of the membranes' market price and increase in their application in the treatment of diverse wastewater. Subsequently, the UF, NF, and RO membranes have gained a primacy in water treatment [15]. The treatment of organic compounds through the UF, NF, and RO membranes is carried out primarily by the filtration process [16]. The retention of the polymeric pollutants by the UF, NF, and RO membranes has been frequently affected by the physio-chemical parameters of the system and for the ionized organic compounds, the exclusion of charge has an important role in the stable retention. The size exclusion parameter of the solute retention is the dependence of the size of the soluble compounds and the parameters of the size of the membrane pores. These parameters are frequently anticipated from the manufacturer's molecular weight cut-off (MWCO) data [17].

A high concentration of pollutants can lower down the membrane's performance either due to their accumulation over the membrane surface or due to their deposition within the pores causing them to constrict. In this context, pollutants could be of two types: (a) the pollutants which could bring about damage to the membrane; and (b) those resulting in membrane fouling by increased resistance to membrane separation and decreased membrane productivity and selectivity. One of the strategies for controlling membrane fouling is the application of activated carbon (AC) pretreatment. However, the light weight of AC and the prolonged duration required for its settling could reduce membrane productivity. In these circumstances, magnetic activated carbon (MAC) could be a more suitable pretreatment alternative as, due to its magnetic force and enhanced coagulation properties, it could effectively convert the soluble compounds to coagulated flocs [18,19].

In the present study, two adsorbents, AC and MAC, have been developed from the biomass of *Dalbergia sissoo* in the form of sawdust in an attempt to use this pretreated material for the removal of AZM from aqueous solution, and to control the fouling of membrane caused by the antibiotic AZM.

## 2. Materials and Methods

Azithromycin antibiotic compound with a purity of 99%, chemical formula ($C_{38}H_{72}N_2O_{12}$), molecular weight 749 g/mol and maximum wavelength of absorbance ($\lambda$max) = 208 nm was gained from the industry of Med-Craft (Hayat Abad Peshawar, Pakistan) and applied no additional purification. Its chemical structure is given in Figure 1.

### 2.1. Preparation of Adsorbents

AC and MAC were prepared from sawdust of *Dalbergia Sissoo*. An electric furnace having an in-built inlet (for $N_2$ entry) and outlet (for exhaust gases) was used for the preparation of AC from dried sawdust. It was washed in 0.1 M HCl and double-distilled water and then dried at 70 °C in an electric oven. Likewise, to prepare MAC, the sawdust was mixed well with $FeCl_3.7H_2O$ and $FeSO_4.7H_2O$ (2:1 ratio) with the continuous supplementation of 0.1 M NaOH at 70 °C. The product was charred in the electrical furnace.

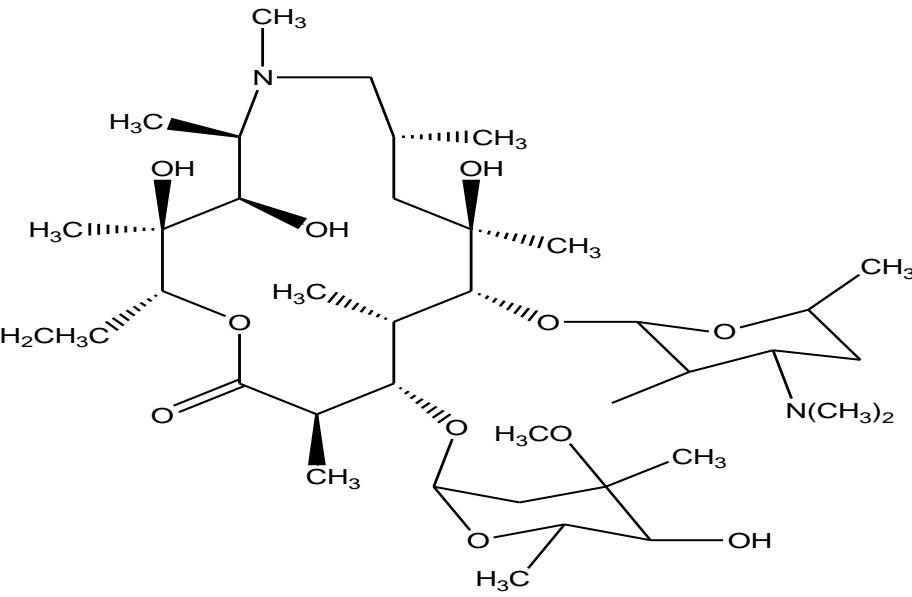

**Figure 1.** Chemical structure of Azithromycin.

*2.2. Characterization*

The composition of elements of AC and MAC was determined through EDS X-sight apparatus. A scanning electron microscope (SEM) at a fixed voltage of 10 kV was employed for observing their surface morphology. For the analyses of represented functional groups, Fourier Transform Infrared (Perkin Elmer, Waltham, MA, USA) was used. Behavior of the prepared adsorbents towards changes in temperature was detected by thermo gravimetric and differential thermal analysis (TG/DTA), applying a diamond series analyzer (Perkin Elmer, Waltham, MA, USA). X-ray diffraction (XRD) analysis was carried out by JOEL X-ray diffractometer JDX-3532, having an Ni filter with Cu Kα radiation with a wavelength of 1.5418°, a voltage of 40 kV, and a current of 30 mA. A Quantachrome analyzer (NOVA, 2200e, Boynton Beach, Florida, USA) was used for determining surface area and pore distribution of adsorbents.

*2.3. Batch Adsorption Experiments*

Stock mixtures of AZM (500 mg/L) were made into water being double-distilled. Working standards were prepared through serial dilutions. Batch tests on adsorption were performed using AZM solution of different concentrations at specified pH, contact time, and temperature. In each case, the temperature-controlled shaker was rotated at 170 rpm for specific intervals. The solution's pH was regulated using 0.1 M of HCl and 0.1 M of NaOH. Optimization experiments were carried out in an initial AZM concentration (10–160 mg/L), contact time (5–240 min), at varied AC and MAC quantities (0.01 to 0.15 g) and at varied pHs (2–12) to attain the equilibrium data. All batch tests were performed in triplicate and means were calculated. After equilibrium attainment, the solution was filtered through Whatman® Grade 42 filter (Rajasthan, India). Post-adsorption concentration of AZM in the solution was recorded at 208 nm. The % removal of AZM along with $q_e$ in mg/g were calculated as follows [20,21]:

$$q_e = (C_i - C_f) \times \frac{V}{m} \tag{1}$$

$$\%R = \frac{(C_i - C_f)}{C_i} \times 100 \tag{2}$$

The $C_i$ and $C_f$ = start and end AZM concentrations, respectively; V = volume of AZM solution (L), and m represents the mass of adsorbents in g.

### 2.4. Isothermal Study

The AZM concentration effect on the AC and MAC adsorption was researched by differentials of the initial AZM concentration in the values of 10–160 mg/L. Other conditions such as optimum volume of solutions of 50 mL, pH, optimum reaction time, and adsorbent dosage 0.1 g were kept constant. Their $q_e$ values were calculated from the Equation (1) and displayed against the concentration at equilibrium. Temkin, Freundlich, Langmuir, Jovanovic and Harkin–Jura models were used to investigate the equilibrium data of adsorption.

### 2.5. Kinetic Study

To determine the kinetic measures of the AZM adsorption on the prepared adsorbents, 0.1 g of AC and MAC were added to 50 mL AZM solution of two different concentrations (80 and 40 mg/L). The solutions were mixed with the respective adsorbent at 170 rpm for 5–240 min. For investigation of the kinetics data, power function, pseudo-first and second-order, Khalaf, Natarajan, and intra-particle diffusion models were used.

### 2.6. pH and Adsorbent Dosage Effect

The adsorption mechanism of the AZM on AC and MAC was studied in range pH from 2 to 12. pH was regulated using HCl and/or NaOH (both 0.1 M). The adsorbent dosage effect was analyzed from 0.01 to 0.15 g of adsorbent dosage. Other optimum parameters remained constant for the pH and adsorbent dosage effect determination.

### 2.7. Effect of Temperature

AZM solution (50 mL) was incorporated with 0.1 g of the prepared adsorbents. The introduction was repeated at several different temperatures in the range 298–333 K. Quantity of AZM adsorbed was measured using Equation (1).

### 2.8. Removal of AZM by Membranes System

UF, NF and RO membranes were employed to measure the removal of AZM from wastewater. Moreover, the membrane permeate flux and % retention was estimated. The characteristics of these membranes are listed in Table 1.

**Table 1.** Characteristic properties of UF, NF and RO membranes.

| UF Membrane | | NF Membrane (Dow Film Tech 2.5 × 40) | | RO Membrane (Dow Film Tech ECO PRO 400 i) | |
|---|---|---|---|---|---|
| **Parameters** | **Specification** | **Parameters** | **Specification** | **Parameters** | **Specification** |
| Membrane type | Capillary multi bore × 7 | model | NF(270–2540) | model | RO(270–2540) |
| Surface area | 50 m$^2$ | Surface area | 3.2 m$^2$ | Surface area | 3.2 m$^2$ |
| Maximum pressure | 109 psi | Maximum pressure | 100–1000 psi | Maximum pressure | 100–1000 psi |
| Membrane back wash pressure | 0.5–1 psi | Membrane back wash pressure | 50–800 psi | Membrane back wash pressure | 50–800 psi |
| MWCO | 100 KD | MWCO | 200–300 | MWCO | 200 |
| Stabilized salt rejection | 10–20% | Stabilized salt rejection | >97% | Stabilized salt rejection | 99.5% |
| Maximum temperature | 40 °C | Maximum temperature | 40–180 °C | Maximum temperature | 40–180 °C |
| pH operating range | 3–10 | pH operating range | 3–10 | pH operating range | 3–10 |
| Backwash pH range | 1–13 | Back wash pH range | 1–12 | Backwash pH range | 1–12 |
| Disinfection chemicals | Hypochlorite and Hydrogen peroxide | Disinfection chemicals | Hydrogen peroxide and peracetic acid | Disinfection chemicals | Hydrogen peroxide and peracetic acid |
| Pore size | 5–20 nm | Pore size | 5–20 nm | Pore size | 5–20 nm |
| Material | Polyethersulfone | Applied pressure | 4.8 bar | Membrane type | Thin film composite |

The membranes were used after washing them with double-distilled water following the instructions of the manufacturer. The samples were kept at neutral pH, room temperature, and at the transmembrane pressure of 1 bar in the course of filtration. The concentration of AZM at the inlet and outlet of the membranes was recorded using UV/Visible spectrophotometer and the percent retention (%R) of AZM was determined from the difference among the two concentrations using the formula [22]:

$$\% \, R = \left(1 - \frac{C_p}{C_b}\right) \times 100 \tag{3}$$

where $C_p$ = and AZM concentration in the permeate, $C_b$ = AZM concentration in the bulk. Likewise, Membrane flux (J) was measured as:

$$J = \frac{1}{A} \times \frac{dv}{dt} \tag{4}$$

where A = membrane's surface area in $m^2$, V = volume (in L) of the permeate, and t = time (in h).

### 2.9. Pilot Plant Studies/Membranes Hybrid Technology

In the pilot plant (Figure 2), both adsorption and membrane filtration processes were integrated to mitigate membrane fouling. Pretreatment with the adsorbents (AC and MAC) was performed in a reactor that was linked to a membrane system. To ensure thorough adsorption before the membrane treatment, each of the adsorbents was separately mixed with AZM solution 2 h before feeding to the three membranes viz., UF, NF, and RO. The UF treatment was done in a dead-end mode, whereas the cross-flow mode was followed for the other two membranes i.e., NF and RO. Pretreatment with MAC was carried on a specifically designed electromagnetic assembly. J and % R were measured for AC and MAC.

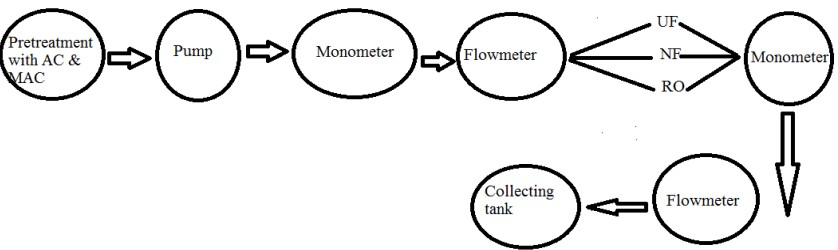

**Figure 2.** Schematic diagram of pilot plant.

## 3. Result and Discussion

### 3.1. Characterization of Adsorbents

The elemental analyses of adsorbents (AC and MAC) are shown in Figure 3a,b. The EDX spectra of AC showed a characteristically high peak of carbon. Small peaks of oxygen, calcium, magnesium, and phosphorus were also observed due to impurities, which originated from the raw sawdust. The deposition of $Fe_3O_4$ on the MAC surface resulted in a lower carbon peak as compared to its complement adsorbent AC. Other elements presented in MAC, such as chlorine and oxygen, were also detected along with a lower percentage of calcium.

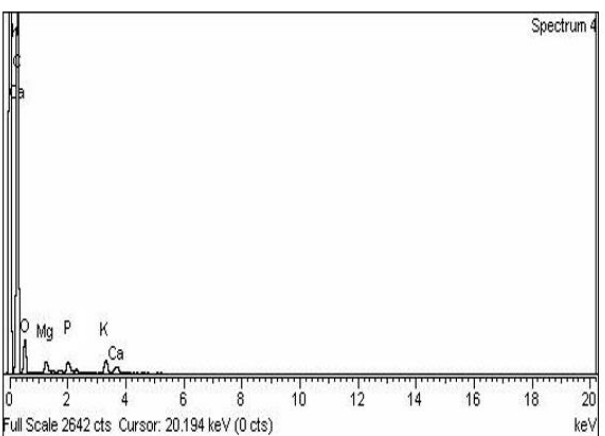 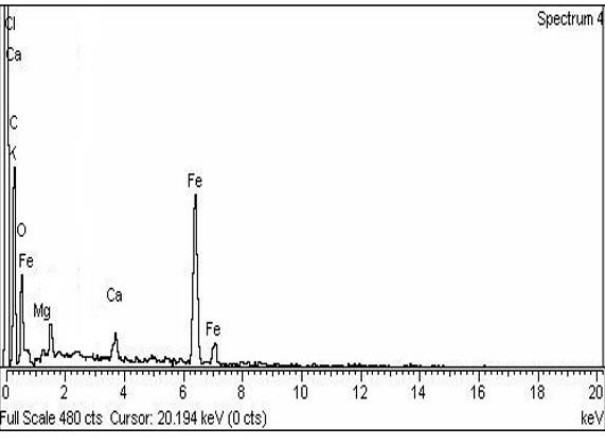

**Figure 3.** EDX spectra of (**a**) AC (**b**) MAC.

The SEM images of AC as represented in Figure 4a,b manifest non-uniform-sized particles, having uneven and distorted edges. The deposited $Fe_3O_4$ could be visualized as spots on MAC's surface in the SEM images (Figure 4c,d).

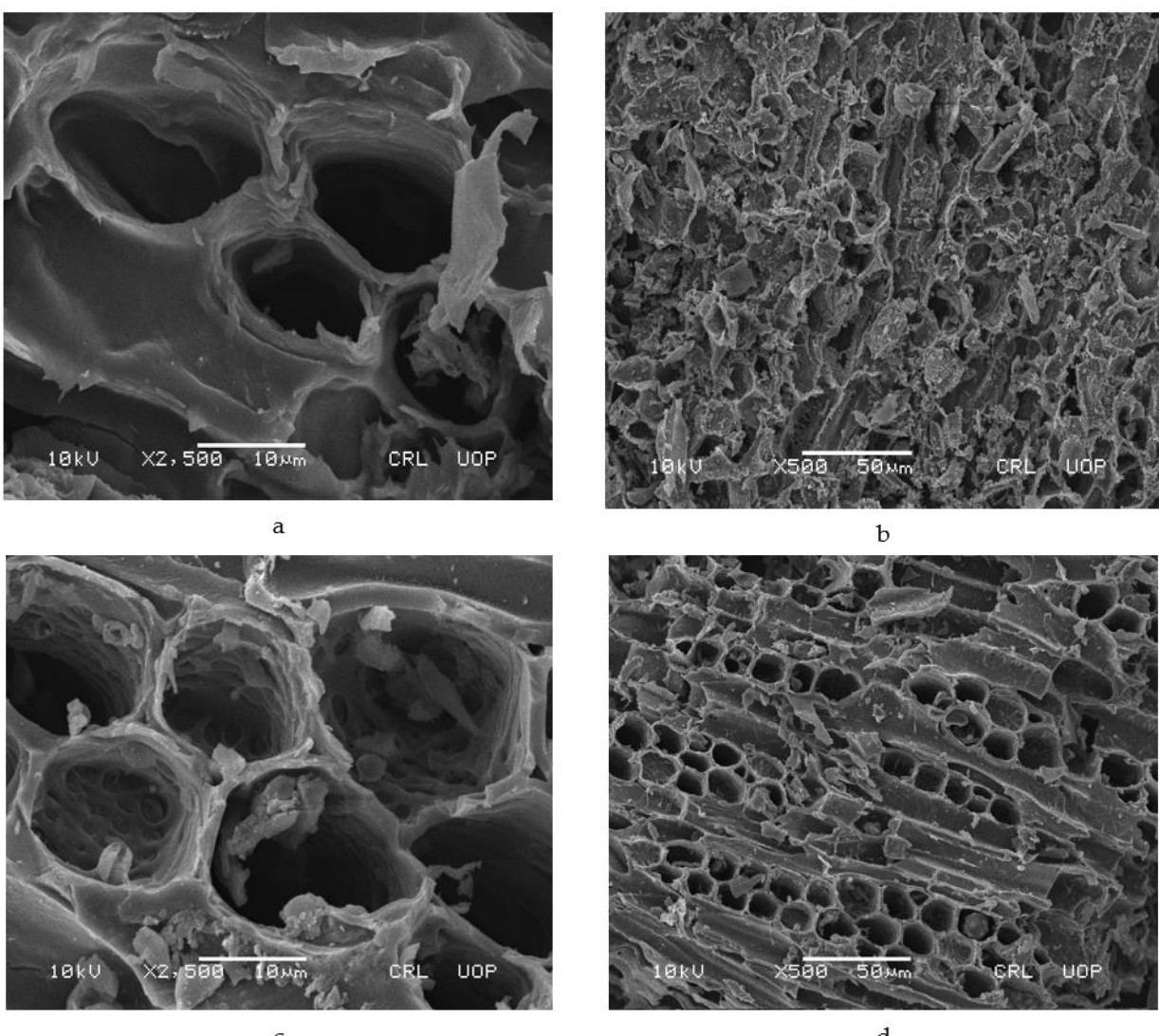

**Figure 4.** SEM images of (**a**) and (**b**) activated carbon, (**c**) and (**d**) magnetic activated carbon.

Figure 5, represent FTIR Spectra of AC and MAC, respectively. A C≡C stretching could be detected from a peak present in the range of 2250–2300 cm$^{-1}$, whereas the presence of tertiary amine is indicated by a peak in the range 3200–3400 cm$^{-1}$ which is considered to be responsible for the binding to AZM to adsorbate surface [23,24]. A peak in the range of 1700–1800 cm$^{-1}$ confirmed the presence of anhydride and aldehydes whereas the peak at 700 cm$^{-1}$ established the presence of iron oxide in MAC [25–27]. There is a noticeable difference in the positions of peaks between the spectra of AC and MAC after AZM adsorption, as evident from Figure 5.

The XRD spectrum of AC (Figure 6) indicates the adsorbent to be amorphous, whereas, according to XRD spectrum of MAC (Figure 6), it is crystalline in nature. Several peaks were observed such as 2θ: 29 (index: 220), 36 (index: 311), 38 (index: 400), and 48 (index: 422) that suggest the presence of magnetite in the composite. Out of the numerous iron oxides present, maghemite and magnetite were deemed to be of special value because of their magnetic properties and their subsequent significance from the environmental perspective [28].

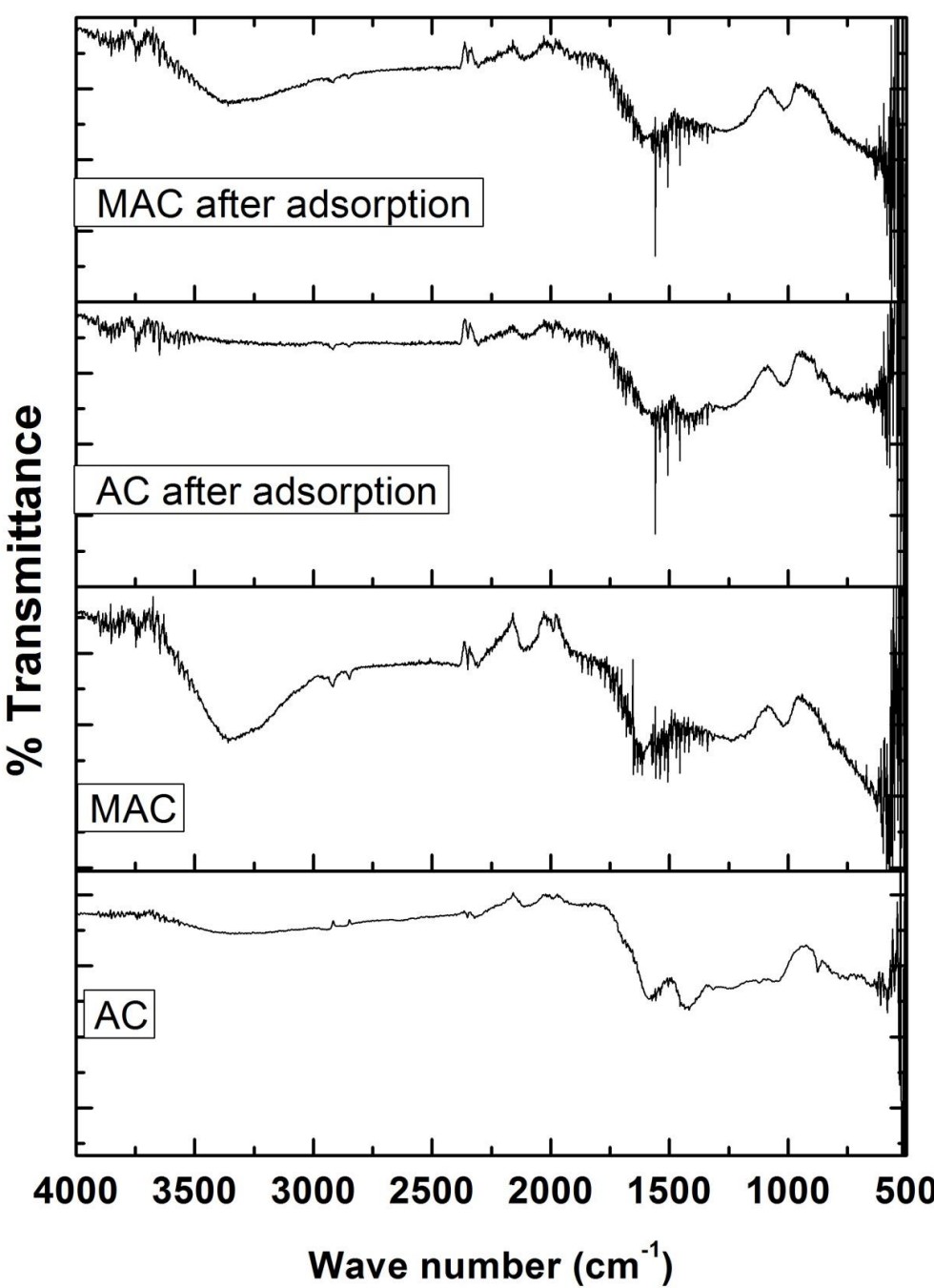

**Figure 5.** FTIR spectra of AC and MAC before adsorption and after adsorption.

According to the TG/DTA spectrum of AC (Figure 7a), at 60–450 °C, there is a net 60% loss of material initially at the first part of the spectrum, then a second 50.35% loss at the next part of the spectrum, and finally a third net loss (33.99%) of the material noticed in the last part of the spectrum. The initial loss is ascribed to the dehydration, the second loss to degradation of cellulose and hemi-cellulose, and the third to the formation of carbon and ash. Likewise, TG/DTA analysis of MAC (Figure 7b), shows an initial 3% loss at 0 to 60 °C, a second 7% loss at 60–318 °C, and a final third loss (58%) that occurred above 318 °C. The initial loss may be attributed to water evaporation, the second to cellulose's decomposition, and the third to the formation of carbonaceous residue due to the transition of $Fe_3O_4$ to FeO.

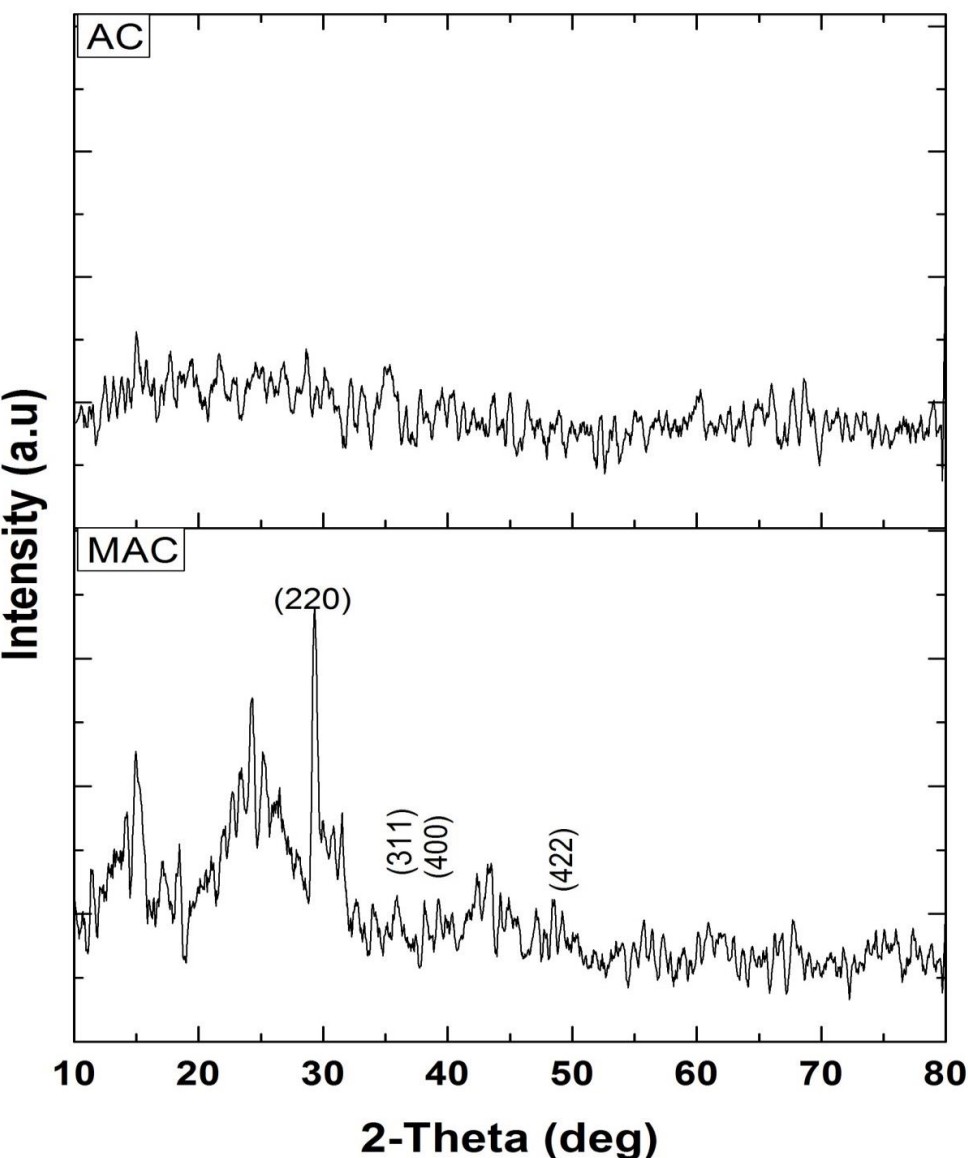

**Figure 6.** XRD spectrum of AC and MAC.

The nitrogen adsorption desorption isotherms (Figure S1a,b) and BJH plots of AC and MAC (Figure S1c,d) suggest that AC has a higher surface area than that of MAC (49.53 and 18.63 m²/g, respectively), bearing in mind the impregnation of $Fe_3O_4$ on the surface of MAC that has lowered the carbon contents per unit volume compared to that of AC. The pore volumes of AC and MAC were observed as 0.019 and 0.016 cm³/g, respectively; likewise, the pore diameters of AC and MAC were detected as 16.11 and 14.85 Å respectively.

*3.2. Isothermal Studies*

A wide range of concentration (10 to 160 mg/L) of AZM was used to study its adsorption upon the prepared adsorbents viz., AC and MAC (Figure 8a). A common trend of increase in adsorption with rise in concentration was observed. To explain the observed adsorption pattern, isotherms such as those of Jovanovic, Langmuir, Temkin, Freundlich, and Harkins–Jura were used. Earlier, efficient removal of tetracycline has been observed with a strong correlation emphasizing Freundlich's adsorption behavior [29]. Therefore, other models were also studied to see the best model characterizing the process.

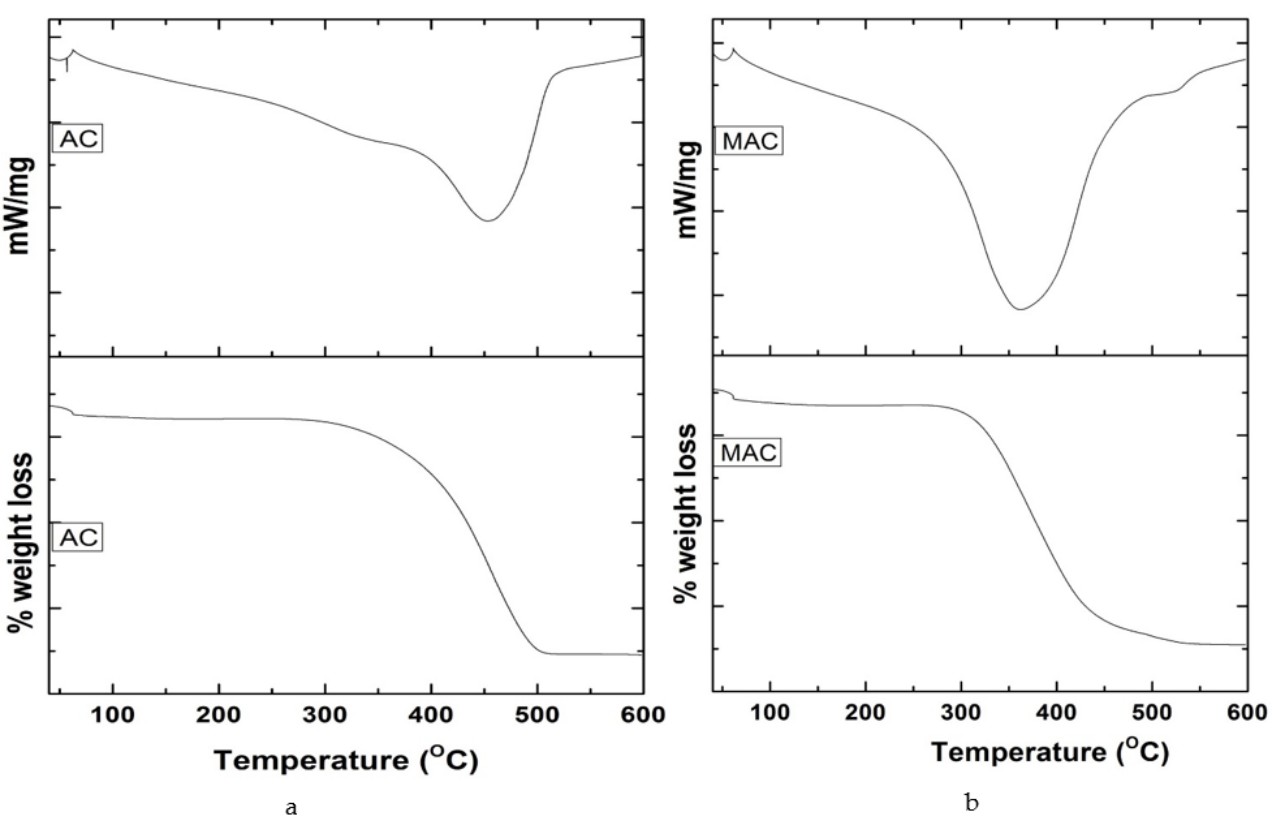

**Figure 7.** TG/DTA analysis of (**a**) AC and (**b**) MAC.

### 3.2.1. Langmuir Isotherm Model

The basic postulate of the Langmuir isotherm model is that adsorption takes place at definite sites, which are equally dispersed throughout the surface of adsorbent. The model in the linear form is represented as Equation (5) [30]:

$$\frac{C_e}{q_e} = \frac{1}{K_L q_m} + \frac{C_e}{q_m} \tag{5}$$

where $q_e$ stands for AZM amount adsorbed at equilibrium (mg/g), $C_e$ for AZM concentration at equilibrium (mg/L), $q_m$ represent maximum uptake ability of AZM by AC and MAC (mg/g), while $K_L$ is a constant related to biosorption energy. Figure 8b represents a plot of $C_e/q_e$ against $C_e$ of AZM for AC and MAC. $q_m$ and $K_L$ were calculated from the slope and intercept of the given plot (Table 2).

### 3.2.2. Freundlich Isotherm Model

A heterogeneous biosorption is suggested by the isotherm upon the surface and active sites with an unequal energy derived for multilayer adsorption. The Freundlich isotherm is represented in Equation (6) [31] as:

$$\ln q_e = \log K_F + \frac{1}{n} \ln C_e \tag{6}$$

where the $K_F$ Freundlich constant stands for adsorption potential and n is empirical constant associated with adsorption strength. Figure 8c illustrates plot of $\ln C_e$ against $\ln q_e$. The value of $K_F$ and n for AC and MAC were calculated from slopes and the intercepts from respective graph. These values are reported in the Table 2.

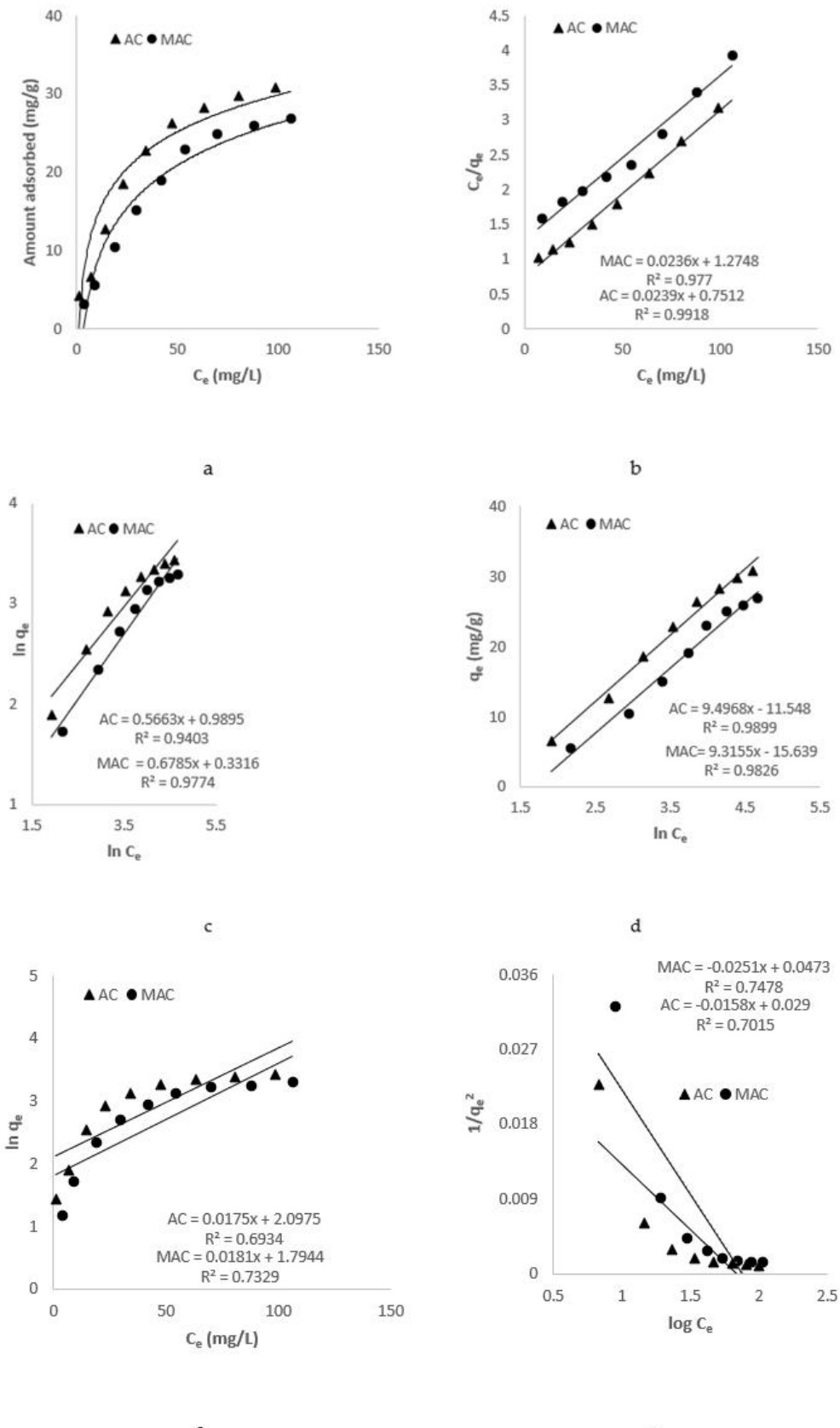

**Figure 8.** AZM adsorption isotherms. (**a**) Effect of concentration on adsorption of AC and MAC (**b**) Langmuir plot (**c**) Freundlich plot (**d**) Temkin plot (**e**) Jovanovich plot (**f**) Harkins–Jura plot.

**Table 2.** Isothermal parameters for adsorption of AZM on AC and MAC.

| Isotherm Models | Parameters | Adsorbents | |
|---|---|---|---|
| | | AC | MAC |
| Langmuir | $q_{max}$ (mg/g) | 41.841 | 42.372 |
| | $K_L$ (L/mg) | 0.0318 | 0.0185 |
| | $R^2$ | 0.9918 | 0.977 |
| Freundlich | $K_F$ (mg/g) | 2.69 | 1.393 |
| | $1/n$ | 0.5663 | 0.6785 |
| | $R^2$ | 0.9403 | 0.9774 |
| Temkin | $\beta$ | 9.4968 | 9.3155 |
| | $\alpha$ | 3.374 | 5.36 |
| | b | 238.998 | 243.65 |
| | $R^2$ | 0.9899 | 0.9826 |
| Jovanovich | $K_J$ (L/g) | 0.0175 | 0.0181 |
| | $q_{max}$(mg/g) | 8.121 | 6.015 |
| | $R^2$ | 0.6934 | 0.7329 |
| Harkins–Jura | $A_H$ ($g^2$/L) | 0.0633 | 0.0399 |
| | $B_H$ ($mg^2$/L) | 1.835 | 1.884 |
| | $R^2$ | 0.7015 | 0.7478 |

### 3.2.3. Temkin Isotherm Model

The Temkin isotherm model assumes that during adsorption process of adsorbate on adsorbent, the free energy of process was associated with surface coverage. The isotherm is represented in linear form as [32]:

$$q_e = \beta \ln \alpha + \beta \ln C_e \tag{7}$$

where: $\beta = RT/b$, R is an ideal gas constant (8.314 J/molK), T is the absolute temperature (K), b represents the adsorption heat constant. Drawing $q_e$ against $\ln C_e$, for AZM by AC and MAC the slope and intercept values were estimated as displayed in Figure 8d and are given in Table 2.

### 3.2.4. Jovanovic Isotherm Model

The Jovanovic model follows the same assumption forwarded by the Langmuir isotherm, with the addition that, while there is absorption, there are mechanical interactions. The isotherm is given in Equation (8) [33]:

$$\ln q_e = \ln q_{max} - KJC_e \tag{8}$$

Here, $C_e$ represents the adsorbate equilibrium concentration (mg/L), $q_e$ represents adsorbate per unit of adsorbent (mg/g), $q_{max}$ shows the maximum adsorption efficiency of adsorbent, and $K_J$ is the Jovanovic constant and is associated with the energy of the AZM adsorption. The linearity of the Jovanovic isotherm was tested by drawing $\ln q_e$ against $C_e$. The value of $K_J$ and $q_{max}$ can be estimated from the slope and intercept of the plot (Figure 8e) and are shown in Table 2.

### 3.2.5. Harkins–Jura Isotherm Model

According to the Harkins–Jura isotherm, the assumption adsorbent surface has a heterogeneous porous structure on which multilayer adsorption of adsorbate occurs. Equation (9) illustrates the isotherm [34].

$$\frac{1}{q_{e^2}} = \frac{B_H}{A_H} - \frac{1}{A_H}\log C_e \tag{9}$$

where B and A are Harkins–Jura constants. The linearity of the isotherm was checked by plotting $1/qe^2$ against log $C_e$, as shown in Figure 8f, and the values of the constant are given in Table 2.

Based on regression-coefficient ($R^2$) values of these isotherm models, it was concluded that the Langmuir isotherm model was the best model that fits the experimental data with high $R^2$ values nearer to 1 and A value nearer to 1, suggest the better curve fitting [35].

### 3.3. Kinetic Study

Kinetic study of AZM adsorption by the two prepared adsorbents (Figure 9a) was carried out at two different concentrations viz., 80 and 40 mg/L. An initial higher rate of adsorption for 30 min was observed following the slower rate up to 120 min. Onwards, it became steady and a constant concentration was reached for both AC and MAC. At 120 min, equilibrium was reached. The observation may be explained as follows. Initially, the active sites for AZM adsorption were free and AZM interacted freely with the active sites for adsorption, and the concentration difference between the AZM solutions and the solid interface was initially higher. This led to a higher rate of adsorption—up to 30 min. Afterwards, AZM were accumulated in the large available surface of the AC and MAC, which lead to occupation of surface binding sites causing adsorption to be slowed down. At last, after 120 min, it became steady, till 240 min duration due to the surface being fully saturated [36]. Various kinetic models, such as Lagergren pseudo-first-order, second-order, power function, Natarajan and Khalaf, and intra-particle diffusion models were applied to calculate the adsorption kinetic parameters.

#### 3.3.1. Pseudo-First-Order Kinetic Model

The mathematical form for the kinetic model of pseudo-first-order is show as [37]:

$$\ln(q_e - q_t) = \ln q_e - K_1 t \tag{10}$$

Here, $q_e$ represent equilibrium amount of AZM (mg g$^{-1}$), $q_t$ represents AZM quantity at instant (t) and $K_1$ represents constant of pseudo-first-order rate. By drawing $\ln(q_e - q_t)$ vs. time (t), $K_1$ and $q_e$ could be deduced from the intercept and slope of the plots, as shown in Figure 9b and Table 3.

**Table 3.** Kinetic characteristics for AZM adsorption of MAC and AC.

| Kinetic Model | Parameters | Adsorbents and Initial Concentrations | | | |
|---|---|---|---|---|---|
| | | AC 80 mg/L | MAC 80 mg/L | AC 40 mg/L | MAC 40 mg/L |
| Pseudo-first-order | $K_1$ (min$^{-1}$) | 0.018 | 0.02 | 0.021 | 0.007 |
| | $q_e$ (mg/g) | 32.727 | 31.705 | 19.492 | 11.271 |
| | $R^2$ | 0.991 | 0.883 | 0.94 | 0.984 |
| Pseudo-second-order | $K_2$(g/mg × min) | $3.97 \times 10^{-4}$ | $1.94 \times 10^{-4}$ | $5.56 \times 10^{-3}$ | $1.43 \times 10^{-4}$ |
| | $q_e$ (mg/g) | 38.61 | 38.76 | 21.097 | 25.51 |
| | $R^2$ | 0.98 | 0.927 | 0.971 | 0.922 |
| Power function | $\alpha$ | 1.124 | 2.121 | 4.968 | 4.74 |
| | b | 0.655 | 0.768 | 0.696 | 0.741 |
| | $R^2$ | 0.931 | 0.969 | 0.95 | 0.976 |
| | $K_N$ (min$^{-1}$) | $5.53 \times 10^{-3}$ | $3.69 \times 10^{-3}$ | $5.76 \times 10^{-3}$ | $3.92 \times 10^{-3}$ |
| | $R^2$ | 0.858 | 0.979 | 0.854 | 0.875 |
| Intra-particle diffusion | $K_{diff}$ (mg/g × min$^{1/2}$) | 2.861 | 2.686 | 1.279 | 1.128 |
| | C | 4.544 | 0.45 | 2.399 | 0.14 |
| | $R^2$ | 0.909 | 0.94 | 0.886 | 0.977 |

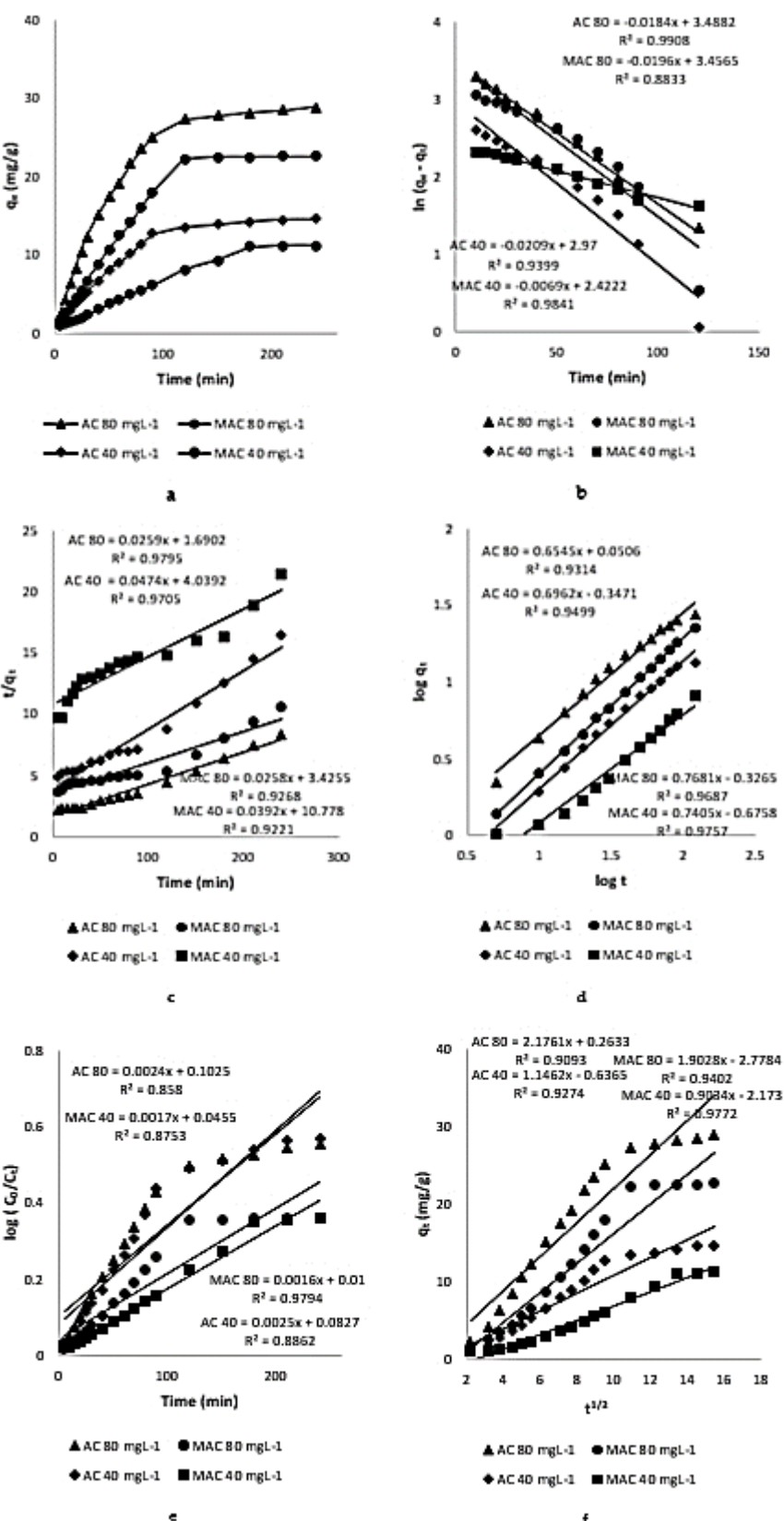

**Figure 9.** Kinetics of adsorption of AZM on AC and MAC (**a**) Kinetic plots (**b**) Plots of kinetics of pseudo-first-order (**c**) Kinetic plots of pseudo-second-order (**d**) Kinetic plots of power function (**e**) Natarajan and Khalaf kinetic plots (**f**) Intra-particle diffusion plots.

### 3.3.2. Pseudo-Second-Order Kinetic Model

The kinetic model of pseudo-second-order in its linear range is given as [37]:

$$\frac{t}{q_t} = \frac{1}{K_2 q_e^2} + \frac{t}{q_e} \tag{11}$$

where, $K_2$ shows the rate constant for pseudo-second-order. By drawing $t/q_t$ against $t$ as depicted in Figure 9c, $k_2$ values were calculated and are given in Table 3.

### 3.3.3. Power Function Kinetic Model

The mathematical equation for the power function kinetic model is given as [38]:

$$\log q_t = \log a + b \log t \tag{12}$$

where $b$ and $a$ are reaction rate constants and their values are obtained from slope and intercept of plots drawn between $\log q_t$ and $\log t$ in Figure 9d.

### 3.3.4. Natarajan and Khalaf Kinetic Model

The linear form of the Natarajan and Khalaf model is expressed as [39]:

$$\log\left(\frac{C}{C_t}\right) = \frac{K_N}{2.303} t \tag{13}$$

In the equation, $C_o$ ($mgL^{-1}$) is the initial concentration when time is zero, $C_t$ (mg/L) shows the concentration of metal ions at instant $t$. The Figure 9e of $\log (C_o/C_t)$ and $t$ gives the value of $K_N$.

### 3.3.5. Intraparticle Diffusion Model

The linearized form of the intra-particle diffusion model is shown as [40]:

$$q_t = K_{diff} t^{1/2} + C \tag{14}$$

where $q_t$ is the quantity of adsorbate adsorbed on the adsorbent at any time; plotting $q_t$ against $t^{1/2}$ (Figure 9f), the values of slope ($K_{diff}$) and intercept $C$ can be estimated.

Based on $R^2$ values of these kinetic models, it was concluded that the pseudo-second-order model was best model that accommodated the experimental data with high regression-constant values.

### 3.4. Effect of pH

Figure 10a represents the pattern of adsorption by AC and MAC with the varying pH of the solution. The adsorption of AZM on AC and MAC was high at neutral pH. The changes in the AZM adsorption capacity of AC and MAC under different solutions' pHs were almost the same. Maximum adsorption of AZM adsorption by AC was achieved at pH = 7 while for MAC at pH = 6, respectively. A speciation graph of AZM (Figure 10b) supports the observed changes in pH with varying pH as a unionized form more soluble in water, and the pKa value is round about 8.5 [41]. At acidic pH it is neutral, while near neutral or slightly above neutral pH is its ionized form, where positive interactions with adsorbent surface groups leads to enhanced adsorption rates in the case of both adsorbents.

### 3.5. Effect of Adsorbents Dosage

The effect of adsorbents' dosage, such as AC and MAC on the AZM adsorption, was studied in the range of 0.01–0.15 g, as shown in Figure 10c. It was clear that adsorption of AZM increased by increasing the dosage of AC and MAC up to 0.1 g. Further increase from 0.1 to 0.15 g resulted no significant change in the AZM adsorption, as also shown before [42]. Therefore, 0.1 g of AC and MAC was considered as an optimum dosage for all the batch adsorption experiments.

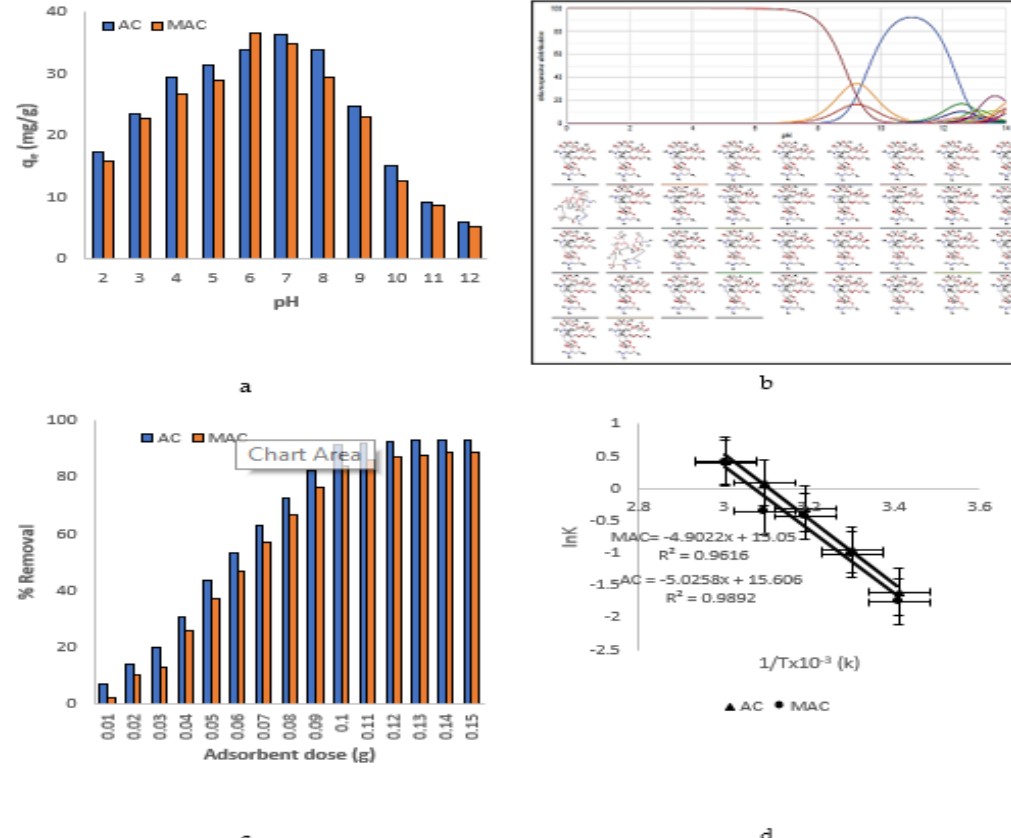

**Figure 10.** (**a**) Effect of pH (**b**) microspecies distribution (**c**) effect of adsorbents dose (**d**) Van't Hoff plots.

### 3.6. Thermodynamic Analysis

Van't Hoff's equation was used to determine the standard enthalpy change ($\Delta H$) and entropy change ($\Delta S$) for the AZM adsorption on AC and MAC. Van't Hoff's equation in linearized form is given as [43]:

$$\ln K = \frac{\Delta S}{R} - \frac{\Delta H}{RT} \tag{15}$$

where $k = q_e/C_e$ and stands for the adsorption affinity. R is universal gas constant with a value of 8.314 (J/mol, K), and T is the temperature in K. The values of the $\Delta H$ and $\Delta S$ were calculated by plotting lnK versus 1/T as shown in Figure 10d and are given in Table 4. The values of $\Delta H°$ were negative and for $\Delta S$ the values were positive, pointing towards the exothermic and spontaneous nature of the AZM adsorption by AC and MAC. $\Delta G$ at different temperatures was calculated as [44,45]:

$$\Delta G = \Delta H - Y(\Delta S) \tag{16}$$

**Table 4.** Thermodynamic parameters for the adsorption of AZM on AC and MAC.

| Adsorbents | $\Delta H$ (KJ/mol) | $\Delta S$ (J/mol·K) | $\Delta G$ (KJ/mol) | | | | |
|---|---|---|---|---|---|---|---|
| | | | 298 K | 303 K | 313 K | 323 K | 333 K |
| AC | $-26.507$ | 91.812 | $-37.974$ | $-39.271$ | $-40.569$ | $-41.909$ | $-43.164$ |
| MAC | $-40.756$ | 125.126 | $-37.500$ | $-37.872$ | $-39.123$ | 40.376 | $-41.625$ |

The values of ΔG < 0 (Table 4) indicates the favorability of the AZM adsorption by AC and MAC [46].

### 3.7. Comparison of Adsorption Capacities of Present Adsorbents with Those Reported in Literature

A comparison has been given in Table 5 as follow. The adsorption capacities of present adsorbents are comparatively high than the reported one presented in table.

**Table 5.** Comparison of adsorption capacities of present adsorbents with those reported in literature.

| Adsorbent | Adsorbate | Qmax (mg/g) | Reference |
|---|---|---|---|
| Activated carbon prepared from *Dilbergia Sissoo* | Azithromycin | 41.841 | This study |
| Activated carbon prepared from biomass | Amoxicillin | 4.4 | [47] |
| Activated carbon prepared from biomass | Sulfonamide | 31 | [48] |
| Activated carbon prepared from *Saccharomyces cerevisiae* | Amoxicillin | 6.27 | [49] |
| Magnetic activated carbon prepared from *Dilbergia Sissoo* | Azithromycin | 42.372 | This study |
| Activated carbon with magnetic $Fe_3O_4$ | Ceftriaxone | 28.93 | [50] |
| Magnetically modified graphene | Amoxicillin | 14.1 | [51] |
| $Fe_3O_4$ nanoparticles | Sulfonamide | 10.83 | [52] |

### 3.8. Effect of AZM on Permeate Flux and Their Improvement by AC and MAC through UF, NF and RO

Figure 11a,c,e, represents how AZM could affect the permeate flux of the membranes used i.e., UF, NF, and RO. In the first case, upon the passage of double-distilled water through the three membranes, a decrease in the permeate flux was observed. The observed effect was due to two reasons: (a) membrane resistance, and (b) presence of ions in the water (as proven by the conductance measurement ($6.3 \times 10^{-6}$ S) leads to ionic interaction. A stable permeate flux was then observed on which there was not any significant effect of the varying experimental conditions or time. The pores of the UF membrane, up to a certain extent, were clogged due to the adsorption of AZM that resulted in a low permeate flux, as the high molecular mass of AZM compared to the MWCO of NF and RO membranes led to lower the permeate flux. The improved permeate flux with AC and MAC was compared as shown in Figure 11b,d,f. The improved permeate flux was observed for AZM, when the pilot plant was used in a hybrid manner with AC and MAC. The improved permeate flux for MAC/membranes was greater as compared to AC/membranes. This was due to the accumulation of AC on the membrane surface and caused pores blockage, while MAC was easily removed from the membrane surface by the application of an electromagnet in the pilot plant, leading to an enhancement in permeate flux. The AC had greater adsorption capability as compared to MAC, but fouling dominated over the adsorption capacity.

### 3.9. Percent Retention of AZM by UF, NF and RO Membranes and Adsorption/Membranes Hybrid Technology

The AZM percent retention is presented in Figure 12a,b. Due to the adsorption of AZM on membrane pores and surface, only 11% retention was seen in case of UF membrane, although the lower mass of AZM relative to the MWCO of UF membrane was suggestive of greater retention. Improved retention after pretreatment with AC (56%) and MAC (52%) were observed (Figure 12a). Likewise, an initial retention (without pretreatment) of 94% was observed in case of NF membrane that got improved to 100% after pretreatment with AC and MAC (Figure 12b). Similarly, in the case of the RO membrane, the observed % retention was 100% after reaction with the adsorbents (data not shown).

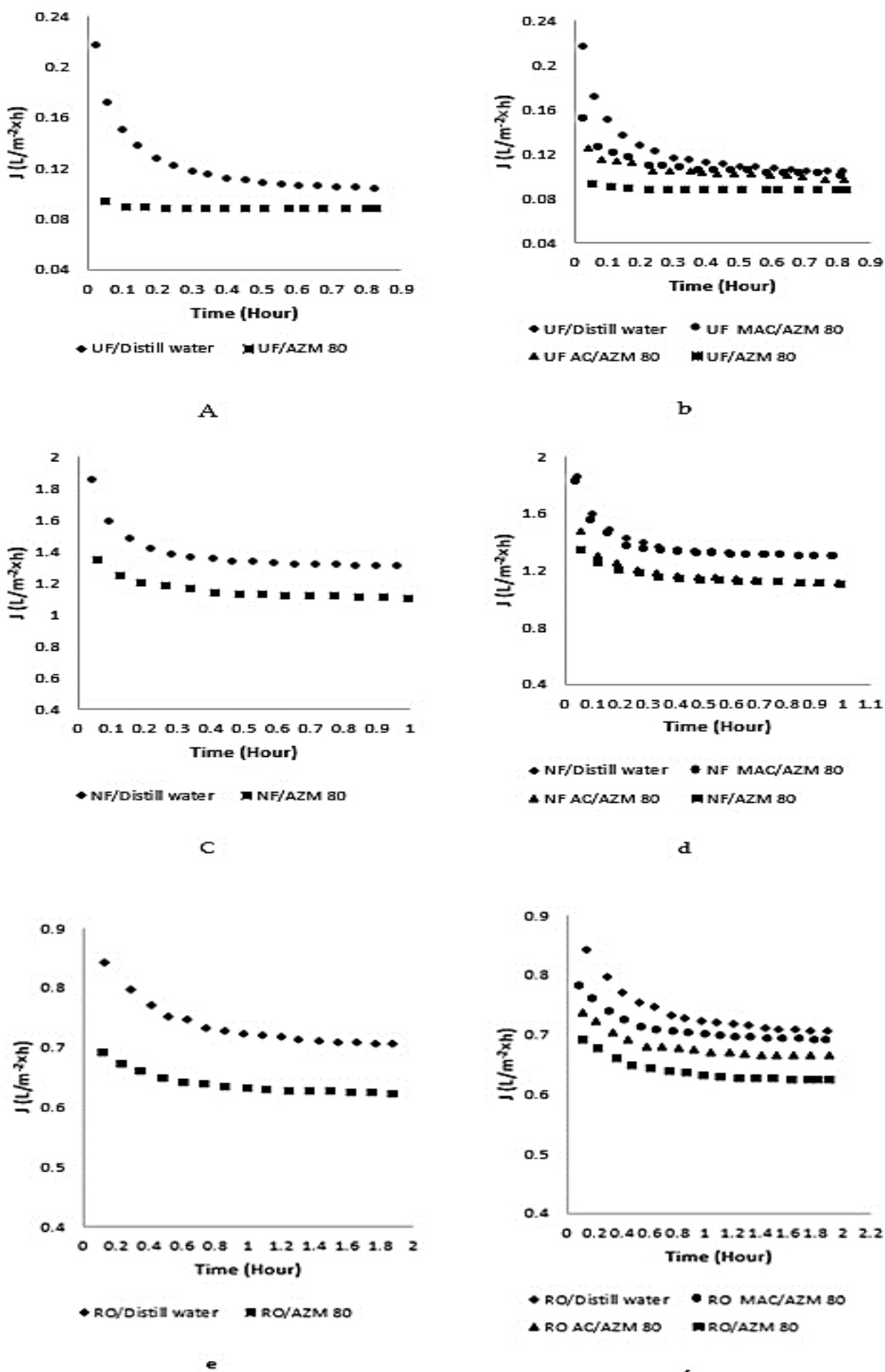

**Figure 11.** Effect of AZM on permeates flux and improvement brought about by AC and MAC (**A**) UF membrane alone (**b**) AC/UF, MAC/UF (**C**) NF membrane alone (**d**) AC/NF, MAC/NF (**e**) RO membrane alone (**f**) AC/RO, MAC/RO.

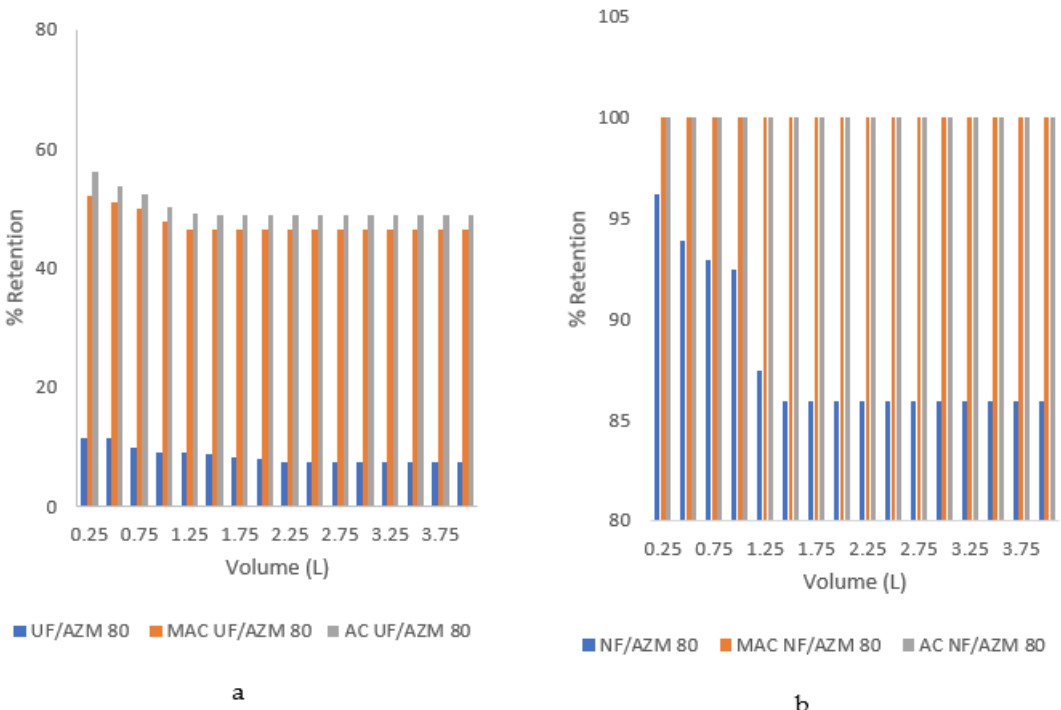

**Figure 12.** Percent retention of AZM by membranes (**a**) UF membrane (**b**) NF membrane.

## 4. Conclusions

AC and MAC were prepared from *Dalbergia Sissoo*, characterized through different methods, such as EDX, XRD, SEM, FTIR surface area analyzer, and TG/DTA and used as efficient adsorbents for avoiding the fouling of the filtration membranes. The Langmuir isotherm model was perfectly applicable for explaining the isotherm data of AC and MAC; kinetic studies indicated the most fit model to be a pseudo-second-order with $R^2$ values ranging from 0.922 to 0.980. AC and MAC pretreatments were compared through membrane hybrid technology, and it was noted that MAC had no membrane-fouling tendency that resulted in improved permeate fluxes, whereas a significant membrane-fouling effect was observed in the case of AC instead of its greater surface area that in turn caused a noticeable decline in the permeate fluxes. Thus, it could be safe to conclude that MAC, because of its magnetic character, would be a better alternative to AC in mitigating membrane fouling.

**Supplementary Materials:** The following are available online at https://www.mdpi.com/article/10.3390/w13141969/s1, Figure S1. N2-adsorption desorption isotherm (a) AC (b) MAC, BJH analysis of (c) AC and (d) MAC.

**Author Contributions:** Conceptualization, M.W.; methodology, M.Z.; software, S.M.S.; validation, I.Z., A.W.K. and M.Z.; formal analysis, A.K., J.B.; investigation, I.Z., A.K., N.P.; resources, M.Z.; data curation, I.Z.; writing—original draft preparation, I.Z. and S.N.; writing—review and editing, I.Z.; visualization, S.M.S.; supervision, M.Z.; project administration, M.Z.; funding acquisition, M.Z. All authors have read and agreed to the published version of the manuscript.

**Funding:** This research was funded by project nr T190087MIMV and European Commission, MLTKT19481R "Identifying best available technologies for decentralized wastewater treatment and resource recovery for India, SLTKT20427 "Sewage sludge treatment from heavy metals, emerging pollutants and recovery of metals by fungi and by project KIK 15392 and 15401 by European Commission.

**Institutional Review Board Statement:** Not applicable.

**Informed Consent Statement:** Not applicable.

**Data Availability Statement:** All the data associated with this publication is presented in this paper. No data is present any other repository to be link here.

**Conflicts of Interest:** The authors declare no conflict of interest. The funders had no role in the design of the study; in the collection, analyses, or interpretation of data; in the writing of the manuscript, or in the decision to publish the results.

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
