# Peer review of "Adsorption-Membrane Hybrid Approach for the Removal of Azithromycin from Water: An Attempt to Minimize Drug Resistance Problem"

_water, doi:10.3390/w13141969_

Round 1

Reviewer 1 Report

Wahab et al. resubmitted this manuscript into Water journal, where the previous manuscript ID was water-1214659. But unfortunately, the reviewer doesnot see any improvement of the resubmitted version and authors havenot addressed the comments from reviewer. So I don’t think this manuscript is suitable for publication and I will keep my previous decision. Further details from my previous comments can be seen below.

Wahab et al. showed the removal of Azithromycin by Activated and Magnetic Activated Carbon, where they have characterized the catalysts with the results of adsorption isotherm models and kinetic models as well as the membrane hybrid technology. However, from the figures and discussion, this work does not meet the requirements of publication. The figure quality in this work is not publishable. Some figures were directly used from raw data generated from machine, which should be redrawn (e.g. FTIR, BET). There is no discussion of isotherm models and kinetic models although authors have shown many figures according to these models. Therefore, this work should be rejected. See following for further details:

  1. The abstract is too lengthy, which should be shortened and summarized with the most important finding.

  1. The organization of Figure 2 should be highly improved. Authors combined all sub-figures together, but they should consider the figure size and how to show the whole combined figure in maximum one page without losing details in all sub-figures.

  1. Page 6 line 199: wording is not precise. How could we find the crystal is in cubic structure from SEM images (Figure 2e and f)?

  1. The analyses of FTIR are not accurate. There is no observable peak in the range of 1700-1800 cm-1 of Figures 2g & 2j. Figure quality is poor, which should be redrawn.

  1. The XRD of figure 2l seems to be strange. The intensity of peaks is quite low and it might be the background instead of diffraction peaks from XRD.

  1. The figures for BET specific surface area are not necessary to show in the manuscript. Instead, N2 adsorption–desorption isotherms and pore size distribution are suggested.

  1. In Langmuir adsorption isotherm equation, author mentioned the meaning of “b” but where is it?

  1. Figure 3: authors showed five different isotherm model but without any discussion of the results and difference. So what is the purpose?

  1. Figure 4 has the same problems. Authors applied different kinetic models but there is no discussion of results.

Author Response

Reviewer 1

Wahab et al. resubmitted this manuscript into Water journal, where the previous manuscript ID was water-1214659. But unfortunately, the reviewer does not see any improvement of the resubmitted version and authors have not addressed the comments from reviewer. So I don’t think this manuscript is suitable for publication and I will keep my previous decision. Further details from my previous comments can be seen below.

 Wahab et al. showed the removal of Azithromycin by Activated and Magnetic Activated Carbon, where they have characterized the catalysts with the results of adsorption isotherm models and kinetic models as well as the membrane hybrid technology. However, from the figures and discussion, this work does not meet the requirements of publication. The figure quality in this work is not publishable. Some figures were directly used from raw data generated from machine, which should be redrawn (e.g. FTIR, BET). There is no discussion of isotherm models and kinetic models although authors have shown many figures according to these models. Therefore, this work should be rejected. See following for further details:

  • Worthy reviewer, the points raised by you are valid and are amendable. Last time the paper was rejected therefore we could not make any changes (as we were not sure to be submitted to same journal). However, this time decision is major revisions and we have tried our best to incorporate all of your suggestions. Hopefully, the manuscript after this revisions will be acceptable to you.

  1. The abstract is too lengthy, which should be shortened and summarized with the most important finding.
  • The abstract size has been reduced accordingly.
  1. The organization of Figure 2 should be highly improved. Authors combined all sub-figures together, but they should consider the figure size and how to show the whole combined figure in maximum one page without losing details in all sub-figures.
  • Worthy reviewer, figure 2 was split into 6 separate figures accordingly.
  1. Page 6 line 199: wording is not precise. How could we find the crystal is in cubic structure from SEM images (Figure 2e and f)?
  • Worthy reviewer, the statement was rephrased accordingly. The word was added as these crystals are cubic otherwise we do not have any procedure to determine it from SEM images.
  1. The analyses of FTIR are not accurate. There is no observable peak in the range of 1700-1800 cm-1 of Figures 2g & 2j. Figure quality is poor, which should be redrawn.

       Ans: The figures of FTIR were redrawn and figure quality has been improved accordingly.

  1. The XRD of figure 2l seems to be strange. The intensity of peaks is quite low and it might be the background instead of diffraction peaks from XRD.

Ans: The XRD of 2k and 2l were redrawn. Hopefully it will be ok now.

  1. The figures for BET specific surface area are not necessary to show in the manuscript. Instead, N2 adsorption–desorption isotherms and pore size distribution are suggested.
  • Worthy reviewer, the BET surface area can be determined from N2 adsorption-desorption isotherm as such there is no BET specific surface area graph. The provided graph is N2 adsorption-desorption isotherm while the second one is about pore volume and distribution etc.
  1. In Langmuir adsorption isotherm equation, author mentioned the meaning of “b” but where is it?
  • Worthy reviewer here it is KL which is a constant and related to adsorption energy. Sometime it is represented by b that’s why the mistake have occurred. It has now been corrected accordingly.
  1. Figure 3: authors showed five different isotherm models but without any discussion of the results and difference. So what is the purpose?
  • We applied different isotherm models in order to determine the best fitting of isothermal data based on R2 The R2 values were greater in case of Langmuir isotherm which were mentioned in the last paragraph of the respective section. Worthy reviewer, you know it better that all research papers are subjected to similarity index checking where if established facts like differences and validities of these isotherms are added would results in high similarity index that would lead to rejection of the paper at editorial stage. Such details are repeatedly reproduced in many papers That is why we have not included these details.
  1. Figure 4 has the same problems. Authors applied different kinetic models but there is no discussion of results.
  • Based on R2 values from these kinetic models we decide the best model where Pseudo-second order model has been found to be the best model that accommodated the experimental data with high regression constant values and have been mentioned in the last paragraph of the section. Worthy reviewer, you know it better that all research papers are subjected to similarity index checking where if established facts like differences and validities of these isotherms if added would results in high similarity index that would lead to rejection of the paper at editorial stage. Such details are repeatedly reproduced in many papers That is why we have not included these details.

Reviewer 2 Report

The authors of the paper developed an adsorption-membrane hybrid system for removing antibiotics in aqua. It must be great if the authors could address the following questions.

  1. The authors mentioned that the introduction of AC or MAC could effectively avoid membranes' fouling resulting in protecting the membrane. It seems that the membrane does not contribute any function of removing AZM. Please clarify the research purpose and motivation in the introduction paragraph.
  2. Line 193 shows the FTIR result for characterizing the functional group of AC and MAC. However, the characterized peaks might have some mistakes. Please re-check the meaning for each FTIR peak.
  3. The functional group of AC and MAC is not proper using FTIR. The main reason is that the AC and MAC are black samples, which would completely absorb light. Therefore, the single would not be clear to distinguish. It is better to use XPS to recognize functional groups.
  4. Line 200 mentioned XRD spectrum. Please cite the reference for XRD results. Morever, the peak information authors mentioned is not corresponding to XRD spectrum. Please check the results.
  5. The figure should be consistent. Figure 2m and 2n were presented in different styles. The description at the range of lines 207 to 216 is not matched with Figure 2m and 2n. Please rewrite this paragraph and cite references for supporting the authors' hypothesis.
  6. The authors used several models for presenting the adsorption behaviors. Based on the correlation coefficient, the authors indicated the Langmuir isotherm is much proper model. Could you compare with other papers and support your result? 
  7. Line 335 presents the microscpcies distribution. Could you explain this figure in detail? It seems to miss some connection between the description and the figure.
  8. Could you explain the contribution of magnetite in this research?
  9. The symbol should be consistent in Figure 6.

Author Response

Reviewer 2:

The authors of the paper developed an adsorption-membrane hybrid system for removing antibiotics in aqua. It must be great if the authors could address the following questions.

  1. The authors mentioned that the introduction of AC or MAC could effectively avoid membranes' fouling resulting in protecting the membrane. It seems that the membrane does not contribute any function of removing AZM. Please clarify the research purpose and motivation in the introduction paragraph.
  • Worthy reviewer, membrane stop them from flowing into permeate flux which is the prime objective of membrane technologies. However, the rejected molecules concentrate near membrane surface and due to concentration polarization problem, they get adsorbed onto membrane surface and block their pores therefore such pretreatment with adsorbents are applied. If pretreatment is with activated carbon it would form a porous layer which would effect the permeate flux, therefore magnetic activated carbon has been used as remedy that was stop from entering into membrane system through application of magnet. From 71 to 80 this fact has already been explained.
    1. Line 193 shows the FTIR result for characterizing the functional group of AC and MAC. However, the characterized peaks might have some mistakes. Please re-check the meaning for each FTIR peak.
  • The FTIR spectra has been redrawn and values haves been corrected accordingly.
    1. The functional group of AC and MAC is not proper using FTIR. The main reason is that the AC and MAC are black samples, which would completely absorb light. Therefore, the single would not be clear to distinguish. It is better to use XPS to recognize functional groups.
  • Worthy reviewer, we do not have the XPS analyzer therefore FTIR analysis has been performed.
    1. Line 200 mentioned XRD spectrum. Please cite the reference for XRD results. Morever, the peak information authors mentioned is not corresponding to XRD spectrum. Please check the results.
  • Reference 23 has accordingly been added and the plots were redrawn with corrected values.
    1. The figure should be consistent. Figure 2m and 2n were presented in different styles. The description at the range of lines 207 to 216 is not matched with Figure 2m and 2n. Please rewrite this paragraph and cite references for supporting the authors' hypothesis.
  • These were redrawn and described accordingly with proper reference.
    1. The authors used several models for presenting the adsorption behaviors. Based on the correlation coefficient, the authors indicated the Langmuir isotherm is much proper model. Could you compare with other papers and support your result? 
  • The results were supported by adding Table 5.
    1. Line 335 presents the microscpcies distribution. Could you explain this figure in detail? It seems to miss some connection between the description and the figure.
  • Worth reviewer, the details about pKa and adsorption has been added accordingly.
    1. Could you explain the contribution of magnetite in this research?
  • Worthy reviewer, there are a number oxides of iron like goethite, webstite, maghemite and magnetite etc. However, among them maghemite and magnetite are important as they are the only two oxides attracted by magnet, the rest are inactive towards magnet. While depositing these oxides on carbon mostly they are converted into goethite therefore, this goal is achieved under inert atmosphere of nitrogen. Our main objective is the magnetic character of the resultant composite to separate and stop it from being entering into membrane system and without maghemite or magnetite this is not possible. I hope the worthy reviewer would understand the its significance in this reaction now.
    1. The symbol should be consistent in Figure 6.
  • Worthy reviewer, there is no such issue with symbols however, in some graphs there are two parameters that in operation of a membrane with distilled water and AZM without added adsorbents while in others there additional two lines for added adsorbents.

Reviewer 3 Report

The manuscript is interesting and suitable for the Journal but before being considered for publication, it requires major revisions. My suggestions are appended below.

1)Line 31-33: The values of ΔG° are not necessary in the abstract. Remove the whole sentence.

2)Line 30: please add the error on the thermodynamic parameters.

3) Line 148: The experiments were carried out in the range of pH 2-12. Are the authors sure that AZM solubility is sufficiently high in the whole range of pH investigated?

4) Section 2.9: A detailed scheme of the Pilot Plant should be provided

5) Eq(10) and Eq(11): The applicability of these equations must be evaluated by non-linear regression analysis. It is well known that linearization of these models frequently leads to erroneous conclusions. (see for example https://doi.org/10.1260/0263-6174.30.3.217)

6) Eq(14): the intraparticle diffusion model cannot be applied to the full set of data, especially to equilibrium data!

7) Tables: all estimated parameters should be accompanied by their associated error.

7) Line 334: The authors say “The rate of adsorption was increased with at pH 6-8.” More information on the effect of pH on the rate of adsorption should be provided, for example by inserting a new graph and/or adding the values of the kinetic parameters.

9) Line 350: The authors are encouraged to carefully check the accuracy of K determination for using in van’t Hoff equation. I suspect that K was not properly calculated (see for more details https://doi.org/10.1016/j.jct.2013.09.013)

10) line 356: the sign of ΔG° does NOT provide information on the spontaneity of the process (see https://doi.org/10.1016/j.molliq.2018.12.019)

Author Response

Reviewer 3:

The manuscript is interesting and suitable for the Journal but before being considered for publication, it requires major revisions. My suggestions are appended below.

1)Line 31-33: The values of ΔG° are not necessary in the abstract. Remove the whole sentence.

  • The sentence has accordingly been deleted.

2)Line 30: please add the error on the thermodynamic parameters.

  • Van’t Hoff plot were redrawn and thermodynamic parameters were corrected. However, if the reviewer wish to put error bars on the graph line we have not performed the experiment in triplicate.

3) Line 148: The experiments were carried out in the range of pH 2-12. Are the authors sure that AZM solubility is sufficiently high in the whole range of pH investigated?

  • Yes its solubility above 8 decreases in water as its pKa value is 8.5 and at/above the ionized form is greatly soluble in lipid rather than water. Below 8 pH its solubility in water is appreciable. Our reaction media is water therefore we needed neutral and acidic pH. That is why our optimum value is round about neutral pH.

4) Section 2.9: A detailed scheme of the Pilot Plant should be provided

  • Pilot Plant scheme was added as figure 2 in the revised paper.

5) Eq(10) and Eq(11): The applicability of these equations must be evaluated by non-linear regression analysis. It is well known that linearization of these models frequently leads to erroneous conclusions. (see for example https://doi.org/10.1260/0263-6174.30.3.217)

  • Worthy reviewer, the suggested reference was added accordingly while as for as the linear and non linear equations are concerned our study about fouling control, the paper is already quite lengthy and addition of such information would further increase the bulk of the data.

6) Eq(14): the intraparticle diffusion model cannot be applied to the full set of data, especially to equilibrium data!

  • Worthy reviewer, it has not been applied to equilibrium data instead it has been applied to kinetics data to evaluate adsorption mechanism.

7) Tables: all estimated parameters should be accompanied by their associated error.

  • Worthy reviewer, the parameters in tables have been estimated from slope and intercept of graphs and you know it better that they single valued for which error estimations are not possible.

7) Line 334: The authors say “The rate of adsorption was increased with at pH 6-8.” More information on the effect of pH on the rate of adsorption should be provided, for example by inserting a new graph and/or adding the values of the kinetic parameters.

  • The detail about this section has been changed. Above 8 pH its ionized is dominant which has low solubility in water therefore neutral and acidic pH adsorptions are high as compared to basic pH as its pKa values is 8.5. kinetics have been provided in a separate section. 

9) Line 350: The authors are encouraged to carefully check the accuracy of K determination for using in van’t Hoff equation. I suspect that K was not properly calculated (see for more details https://doi.org/10.1016/j.jct.2013.09.013)

  • The van’t Hoff equations were redrawn and the thermodynamic parameters were again calculated. Reference was added accordingly.

10) line 356: the sign of ΔG° does NOT provide information on the spontaneity of the process (see https://doi.org/10.1016/j.molliq.2018.12.019).

  • negative∆G means that the reactants, or initial state, have more free energy than the products, or final state. For a spontaneous process initially exothermic nature of reaction was a criteria then entropy and now Gibbs free energy that is in literature. The suggested reference was informative and was incorporated accordingly.

Round 2

Reviewer 1 Report

No further comments.

Author Response

Thank you worthy reviewer

Reviewer 2 Report

The authors of the first revised paper already modified and improved some experimental explanations based on suggestions. But it still some suggestions must be revised.

  1. Line 191, the authors mentioned that C≡C stretching could be detected from a peak range of 2250-2300 cm-1. Do you have any references supporting your explanation? The peaks at this range probably are assigned as CO2 from the background.
  2. The peaks at 1700-1800 cm-1 were assigned as anhydride and aldehydes. But it is difficult to confirm the adsorption of Azithromycin using the peak information you mentioned. The structure of Azithromycin mainly includes ether, alcohol, and ester bonding. Therefore, author's explanation might confirm properly.
  3. To confirm Azothromycin through FTIR, a high concentration of adsorbate is required for sample preparation rather than using the experimental condition author used. It means authors should prepare specific analytic samples for high concentration adsorption using AC and MAC.
  4. The specific structure confirms from FTIR would like to use tertiary amine. This functional group only contributes from the adsorbents. Therefore, this improvement is more reliable and easy to convince readers. 

Author Response

Reviewer 2

The authors of the first revised paper already modified and improved some experimental explanations based on suggestions. But it still some suggestions must be revised.

  1. Line 191, the authors mentioned that C≡C stretching could be detected from a peak range of 2250-2300 cm-1. Do you have any references supporting your explanation? The peaks at this range probably are assigned as CO2 from the background.
  • Respected reviewer, the mentioned range was confirmed from Appendix E: List of Common Vibrational Group. Reference: Frequencies (cm−1) Modern Vibrational Spectroscopy and Micro-Spectroscopy: Theory, Instrumentation and Biomedical Applications, First Edition. Max Diem© 2015 John Wiley & Sons, Ltd. Published 2015 by John Wiley & Sons, Ltd.
  1. The peaks at 1700-1800 cm-1 were assigned as anhydride and aldehydes. But it is difficult to confirm the adsorption of Azithromycin using the peak information you mentioned. The structure of Azithromycin mainly includes ether, alcohol, and ester bonding. Therefore, author's explanation might confirm properly.
  • The literature data suggest that appearance of peaks around the mentioned region indicate the presence of carbonyl band. References ae give as under:
    1. Araujo, J., Ortíz, R., Velásquez, W., & Ortega, J. M. (2006). Determination of Azithromycin in Pharmaceutical Formulations by Differential Pulse Voltammetry: Comparison with Fourier Transformed Infrared Spectroscopic Analysis. Portugaliae Electrochimica Acta24(1), 71-81.
    2. Mallah, M. A., Sherazi, S. T. H., Mahesar, S. A., & Rauf, A. (2011). Assessment of azithromycin in pharmaceutical formulation by fourier-transform infrared (FT-IR) transmission spectroscopy. Pakistan Journal of Analytical & Environmental Chemistry12(1 & 2), 7.
  • As far binding of adsorbate and adsorbent are concerned, on adsorbent surfaces a number of functional groups are present among which few contributes to binding of adsorbate not all of them.
  1. To confirm Azothromycin through FTIR, a high concentration of adsorbate is required for sample preparation rather than using the experimental condition author used. It means authors should prepare specific analytic samples for high concentration adsorption using AC and MAC.
  • Worthy reviewer, the FTIR analysis has been performed for AC and MAC not azithromycin. After adsorption in the spectra only band position has changed due to AZT adsorption. You are right in preparing sample for after adsorption analysis usually high concentrations of adsorbate are used to visualize a clear picture of the adsorption taking place there.
  1. The specific structure confirms from FTIR would like to use tertiary amine. This functional group only contributes from the adsorbents. Therefore, this improvement is more reliable and easy to convince readers. 

The suggestion is acknowledged and highly appreciated.

Reviewer 3 Report

The authors response is not satisfying. In my opinion, the manuscript is not suitable for publication for the reasons explained below:

  • Round 1 Referee comment Point 2)Line 30: please add the error on the thermodynamic parameters.

Authors response: Van’t Hoff plot were redrawn and thermodynamic    parameters were corrected. However, if the reviewer wish to put error   bars on the graph line we have not performed the experiment in       triplicate.

Round 2 Referee comment: The required error is not related to replicates but, obviously, to the curve fitting procedure.

  • Round 1 Referee comment 5) Eq(10) and Eq(11): The applicability of these equations must be evaluated by non-linear regression analysis. It is well known that linearization of these models frequently leads to erroneous conclusions. (see for example https://doi.org/10.1260/0263-6174.30.3.217)

Authors response: Worthy reviewer, the suggested reference was added accordingly while as for as the linear and non linear equations are concerned our study about fouling control, the paper is already quite lengthy and addition of such information would further increase the bulk of the data.

Round 2 Referee’s comment: The linear regression should have been replaced by non linear curve fitting to avoid mistakes.

  • Round 1 Referee comment 6) Eq(14): the intraparticle diffusion model cannot be applied to the full set of data, especially to equilibrium data!

Authors response: Worthy reviewer, it has not been applied to equilibrium data instead it has been applied to kinetics data to evaluate adsorption mechanism.

Round 2 Referee’s comment: As suggested in the first round of revision, the intraparticle model applies only to the early stage of the kinetic process, not certainly to data at equilibrium (or near equilibrium).     

  • Round 1 Referee comment 7) Tables: all estimated parameters should be accompanied by their associated error.

Authors response: Worthy reviewer, the parameters in tables have been estimated from slope and intercept of graphs and you know it better that they single valued for which error estimations are not possible.

Round 2 Referee’s comment: The authors ignore the error calculated from regression analysis.

  • Round 1 Referee comment  Line 334: The authors say “The rate of adsorption was increased with at pH 6-8.” More information on the effect of pH on the rate of adsorption should be provided, for example by inserting a new graph and/or adding the values of the kinetic parameters.

Authors response: The detail about this section has been changed. Above 8 pH its ionized is dominant which has low solubility in water therefore neutral and acidic pH adsorptions are high as compared to basic pH as its pKa values is 8.5. kinetics have been provided in a separate section. 

Round 2 Referee’s comment: The effect of pH on the RATE of adsorption has not been elucidated.

  • Round 1 Referee comment  9) Line 350: The authors are encouraged to carefully check the accuracy of K determination for using in van’t Hoff equation. I suspect that K was not properly calculated (see for more details https://doi.org/10.1016/j.jct.2013.09.013)

Authors response: The van’t Hoff equations were redrawn and the thermodynamic parameters were again calculated. Reference was added accordingly.

Round 2 Referee’s comment: DH° and DS° can be determined from k=qe/Ce only under well-established conditions.

  • Round 1 Referee comment  10) line 356: the sign of ΔG° does NOT provide information on the spontaneity of the process (see https://doi.org/10.1016/j.molliq.2018.12.019).

Authors response: A negative∆G means that the reactants, or initial state, have more free energy than the products, or final state. For a spontaneous process initially exothermic nature of reaction was a criteria then entropy and now Gibbs free energy that is in literature. The suggested reference was informative and was incorporated accordingly.

Round 2 Referee’s comment: Author’s response refers to DG sign. However, in the manuscript their statement refer to DG° whose sign is not related to the spontaneity of the process.  

Author Response

Reviewer 3

The authors response is not satisfying. In my opinion, the manuscript is not suitable for publication for the reasons explained below:

  • Round 1 Referee comment Point 2)Line 30: please add the error on the thermodynamic parameters.

Authors response: Van’t Hoff plot were redrawn and thermodynamic    parameters were corrected. However, if the reviewer wish to put error   bars on the graph line we have not performed the experiment in       triplicate.

Round 2 Referee comment: The required error is not related to replicates but, obviously, to the curve fitting procedure.

  • In the curve error bars were accordingly added.
  • Round 1 Referee comment 5) Eq(10) and Eq(11): The applicability of these equations must be evaluated by non-linear regression analysis. It is well known that linearization of these models frequently leads to erroneous conclusions. (see for example https://doi.org/10.1260/0263-6174.30.3.217)

Authors response: Worthy reviewer, the suggested reference was added accordingly while as for as the linear and non linear equations are concerned, our study is about fouling control, as the paper is already quite lengthy and addition of such information would further increase the bulk of the data.

Round 2 Referee’s comment: The linear regression should have been replaced by non linear curve fitting to avoid mistakes.

  • Worthy reviewer, I could not found non linear curve fitting in Excel, There are certain optional like exponential etc however, no such option non linear option is there to apply. Also once again, it is a lengthy paper and we are concerned about membrane fouling not adsorption. Adsorption in this study is just an application.
  • Round 1 Referee comment 6) Eq(14): the intraparticle diffusion model cannot be applied to the full set of data, especially to equilibrium data!

Authors response: Worthy reviewer, it has not been applied to equilibrium data instead it has been applied to kinetics data to evaluate adsorption mechanism.

Round 2 Referee’s comment: As suggested in the first round of revision, the intraparticle model applies only to the early stage of the kinetic process, not certainly to data at equilibrium (or near equilibrium).  

  • Worthy reviewer, in the provided where I have told that it is applied to equilibrium data. I think you wants that I should follow your ideas as such without caring of my own study. You mean I should write a research paper like your paper. You have some problem that every thing say is correct and no one in the world is not perfect like you. I have not mentioned the world equilibrium anywhere. Also I told the paper is about membrane fouling where adsorption has been used as an application. I think you mean that I should leave all aims of my study and only follow you.
  • Please look into literature if I have made any deviation from literature then tell me. You are behind you own shop.
  • Round 1 Referee comment 7) Tables: all estimated parameters should be accompanied by their associated error.

Authors response: Worthy reviewer, the parameters in tables have been estimated from slope and intercept of graphs and you know it better that they single valued for which error estimations are not possible.

Round 2 Referee’s comment: The authors ignore the error calculated from regression analysis.

  • Worthy reviewer, as I told they are calculated from slope and intercept which are single values. Please give me only one example from literature where they have written in such format you wants.
  • Round 1 Referee comment  Line 334: The authors say “The rate of adsorption was increased with at pH 6-8.” More information on the effect of pH on the rate of adsorption should be provided, for example by inserting a new graph and/or adding the values of the kinetic parameters.

Authors response: The detail about this section has been changed. Above 8 pH its ionized is dominant which has low solubility in water therefore neutral and acidic pH adsorptions are high as compared to basic pH as its pKa values is 8.5. kinetics have been provided in a separate section. 

Round 2 Referee’s comment: The effect of pH on the RATE of adsorption has not been elucidated.

  • All details are there but you have some problem in understanding the writing of others

  • Round 1 Referee comment  9) Line 350: The authors are encouraged to carefully check the accuracy of K determination for using in van’t Hoff equation. I suspect that K was not properly calculated (see for more details https://doi.org/10.1016/j.jct.2013.09.013)

Authors response: The van’t Hoff equations were redrawn and the thermodynamic parameters were again calculated. Reference was added accordingly.

Round 2 Referee’s comment: DH° and DS° can be determined from k=qe/Ce only under well-established conditions.

  • From where I have calculated. Where I have mentioned that I have not determined from it. You are blaming for such error which we have not made or even mentioned.  
  • Round 1 Referee comment  10) line 356: the sign of ΔG° does NOT provide information on the spontaneity of the process (see https://doi.org/10.1016/j.molliq.2018.12.019).

Authors response: A negative∆G means that the reactants, or initial state, have more free energy than the products, or final state. For a spontaneous process initially exothermic nature of reaction was a criteria then entropy and now Gibbs free energy that is in literature. The suggested reference was informative and was incorporated accordingly.

Round 2 Referee’s comment: Author’s response refers to DG sign. However, in the manuscript their statement refer to DG° whose sign is not related to the spontaneity of the process. 

  • The DG was corrected as DG

Round 3

Reviewer 2 Report

Thanks for the authors' response. My questions are already fully answered. Congratulations! 

Author Response

Thank you worthy reviewer, for your positive response

This manuscript is a resubmission of an earlier submission. The following is a list of the peer review reports and author responses from that submission.

Round 1

Reviewer 1 Report

The authors of the paper demonstrated for antibiotic -Azithromycin removal through AC and MAC. It would be great if the authors could address the following suggestion.

  1. The presentation of each figure is not clear and lacks consistency. For example, SEM images would be separated from other material analyses. These figures should be re-arranged.
  2. In addition, the description and explanation of FTIR are not correct. The characteristic peak located at 2250 to 2300cm-1 was assigned as carbon-carbon triple bonding. But this functional group does not exist in your structure. This peak might be CO2 in the atmosphere rather than material.  Authors should follow the chemical structure of your material for FTIR analysis. In addition, the noise is too high resulting in the figure is not proper for publication. The AC and MAC analysis should be processed by XPS.
  3. For BET result, authors should redraw the figure rather than raw data including for publication.
  4. The discussion of absorption just is considered as data analysis. It is necessary for including adsorption mechanism based on different materials. 

Reviewer 2 Report

Manuscript Number: water-1214659

Effective Removal of Azithromycin from Aqueous Solution Through Activated and Magnetic Activated Carbon Prepares from Sawdust of Dalbergia Sissoo and Post-Treatment Pilot Membrane Hybrid Technology.

The present work investigates the performance of combined AC or MAC adsorption-NF/RO/UF for the removal of azithromycin from an aqueous solution. The possibility of using this hybrid system is subsequently tested on a pilot scale. A complete characterization of both adsorbent materials (EDX, SEM, FTIR, XRD, TG / DTA, BET and BJH) is carried out, as well as adsorption tests and adjustment to different kinetic models to find the best adjustment. A comparative analysis of the permeability of the membranes of the simple system (without adsorbent pretreatment) and a hybrid system is carried out comparing different types of membranes in a pilot-scale installation.

GENERAL COMMENTS

The research carried out in this study is very interesting, I consider that an extensive characterization of adsorbent materials has been carried out, a comprehensive treatment of the data obtained, adjusting them to different models. The experimental procedure followed is quite clear and the addition of an intermediate adsorption step prior to filtration for the removal of the pollutant appears to be duly justified, while it is true that I have missed any reference to studies demonstrating membrane fouling justifying the incorporation of adsorption into the process, longer filtrability studies may have shed light on this.

I think the work is interesting and it should be accepted after some minor revisions.

SPECIFIC COMMENTS

(120) There is a duplicate word (using).

Fig. 2 should be separated, figures 2k and 2l should have the same scale and format to improve comparability. Figures 2m and 2n, the lower graph I understand that the abscissa axis is not "Weight loss%". Also, in both cases, the format should match. Figures 2o, 2p, 2q and 2r are irrelevant, they do not provide additional information to what is explained in the text.

(201) The text refers to Figure 2g and 2j, but I think it actually refers to Figure 2g and 2h.

(205-207) The text says that there is no difference in the position of the peaks, however, there are differences in their shape. What is the reason for this phenomenon?

(211) Reference should be made to Figure 2l.

(215-218) The weight loss percentages are not those seen in graph 2m.

(215-224) The first division of graph 2n does coincide with the 60ºC to which the text refers, however, the second division does not coincide with the 318ºC to which the authors refer in this paragraph.

(225) There must be an error in the values of the BET surface, as specified in graphs 2o and 2q, the value of the BET surface for AC and MAC is 41.97 and 49.53 m2/g, respectively, and for the BJH, 18.63 and 21.76 m2/g. Considering the latter, the reasoning carried out in lines 226 to 227 is not understood.

(231) There is a space left at the end of the line.

Table 2. The name Hurkins-Jura should be corrected to Harkins-Jura.

In the text, the tables are named first before the images, in a printed article they should, therefore, appear before. This happens with Tables 2 and 3 that appear before Figures 3 and 4, respectively.

(335-337) If the mean of the R2 values ​​obtained with each of the models is made, the one that gives the best result is the "Power function" model, but the Pseudosecond order model has been considered optimal. What criteria has been chosen for your choice? What are the implications of applying this model?

(340) It increases at values of pH of 6-8.

(345) Considering the previous correction, it does not appear that this is the case.

Fig 5b the resolution should be improved, it is unintelligible.

(362) The units of the constant R are kJ/mol·K.

(376) The units of conductance are Siemens, whose abbreviation according to the International System is S.

Fig. 6 Figures 6a, 6c and 6e repeat information provided in Figures 6b, 6d and 6f. The authors should consider the convenience of improving the comparison in this graph, I think it would be clearer that instead of indicating all the possibilities of one of the technologies, different technologies should be compared with the same configuration.

How do the authors explain a decrease in flux, therefore fouling, in the case of filtration of distilled water?

(382-383) There is no significant improvement in some cases of NF, specifically the hybrid AC/NF system with respect to NF.

Figures 7a and 7b should have the same scale, however, in no case should the elimination percentage be represented on a scale that exceeds 100% (Fig 7b). Fig 7c. It is irrelevant, it could be indicated in the text that the elimination is 100% of all cases.

(385-389) Have more extensive experiments have been done to assert that fouling has occurred? What have they consisted of?

Reviewer 3 Report

Wahab et al. showed the removal of Azithromycin by Activated and Magnetic Activated Carbon, where they have characterized the catalysts with the results of adsorption isotherm models and kinetic models as well as the membrane hybrid technology. However, from the figures and discussion, this work does not meet the requirements of publication. The figure quality in this work is not publishable. Some figures were directly used from raw data generated from machine, which should be redrawn (e.g. FTIR, BET). There is no discussion of isotherm models and kinetic models although authors have shown many figures according to these models. Therefore, this work should be rejected. See following for further details:

  1. The abstract is too lengthy, which should be shortened and summarized with the most important finding.

  1. The organization of Figure 2 should be highly improved. Authors combined all sub-figures together, but they should consider the figure size and how to show the whole combined figure in maximum one page without losing details in all sub-figures.

  1. Page 6 line 199: wording is not precise. How could we find the crystal is in cubic structure from SEM images (Figure 2e and f)?

  1. The analyses of FTIR are not accurate. There is no observable peak in the range of 1700-1800 cm-1 of Figures 2g & 2j. Figure quality is poor, which should be redrawn.

  1. The XRD of figure 2l seems to be strange. The intensity of peaks is quite low and it might be the background instead of diffraction peaks from XRD.

  1. The figures for BET specific surface area are not necessary to show in the manuscript. Instead, N2 adsorption–desorption isotherms and pore size distribution are suggested.

  1. In Langmuir adsorption isotherm equation, author mentioned the meaning of “b” but where is it?

  1. Figure 3: authors showed five different isotherm model but without any discussion of the results and difference. So what is the purpose?

  1. Figure 4 has the same problems. Authors applied different kinetic models but there is no discussion of results.